# Bayesian Quantification with Black-Box Estimators

**Albert Ziegler**                                                                                    *albert@xbow.com*
*XBOW, Head of AI*
*Uppsala, Sweden*

**Paweł Czyż**                                                                          *pawelpiotr.czyz@ai.ethz.ch*
*ETH AI Center and Department of Biosystems Science and Engineering*
*ETH Zürich*
*Zürich, Switzerland*

**Reviewed on OpenReview:** *https: // openreview. net/ forum? id=Ft4kHrOawZ*

## Abstract

Understanding how different classes are distributed in an unlabeled data set is important for the calibration of probabilistic classifiers and uncertainty quantification. Methods like adjusted classify and count, black-box shift estimators, and invariant ratio estimators use an auxiliary and potentially biased black-box classifier trained on a different data set to estimate the class distribution on the current data set and yield asymptotic guarantees under weak assumptions. We demonstrate that these algorithms are closely related to the inference in a particular probabilistic graphical model approximating the assumed ground-truth generative process, and we propose a Bayesian estimator. Then, we discuss an efficient Markov chain Monte Carlo sampling scheme for the introduced model and show an asymptotic consistency guarantee in the large-data limit. We compare the introduced model against the established point estimators in a variety of scenarios, and show it is competitive, and in some cases superior, with the non-Bayesian alternatives.

## 1 Introduction

Consider a medical test predicting illness (classification label $Y$), such as influenza, based on symptoms (features $X$). This often can be modeled as an anti-causal problem[1] (Schölkopf et al., 2012), where $Y$ causally affects $X$. Under the usual i.i.d. assumption, one can approximate the probabilities $P(Y \mid X)$ using a large enough training data set.

However, the performance on real-world data may be lower than expected, due to data shift: the issue that real-world data comes from a different probability distribution than training data. For example, well-calibrated classifiers trained during early stages of the COVID-19 pandemic will underestimate the incidence of the illness at the time of surge in infections.

The paradigmatic case of data shift is *prior probability shift*, where the context (e.g., training and test phase) influences the distribution of the target label $Y$, although the generative mechanism generating $X$ from $Y$ is left unchanged. In other words, $P_{\text{train}}(X \mid Y) = P_{\text{test}}(X \mid Y)$, although $P_{\text{train}}(Y)$ may differ from $P_{\text{test}}(Y)$. If $P_{\text{test}}(Y)$ is known, then $P_{\text{test}}(Y \mid X)$ can be calculated by rescaling $P_{\text{train}}(Y \mid X)$ according to Bayes' theorem (see Saerens et al. (2001, Sec. 2.2) or Schölkopf et al. (2012, Sec. 3.2)). However, $P_{\text{test}}(Y)$ is usually unknown and needs to be estimated having access only to a finite sample from the distribution $P_{\text{test}}(X)$. This task is known as *quantification* (González et al., 2017; Forman, 2008).

Although quantification found applications in adjusting classifier predictions, it is an important problem on its own. For example, imagine an inaccurate but cheap COVID-19 test, which can be taken by a

---

[1] While influenza causes high fever, in many medical problems the causal relationships are much more complex (Castro et al., 2020).

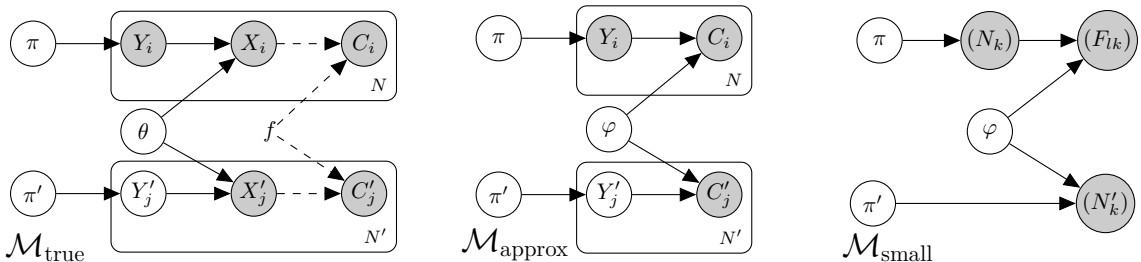

Figure 1: Left: High-dimensional model $\mathcal{M}_{\text{true}}$. Dashed arrows are used to create low-dimensional representations $C_i$ and $C'_j$ using a given black-box function $f$. Filled nodes represent observed random variables. The top row represents the labeled data set and the bottom row represents the unlabeled data set. Middle: Tractable approximation $\mathcal{M}_{\text{approx}}$ (Sec. 3). Right: Model $\mathcal{M}_{\text{small}}$ enabling fast inference when $f$ is a black-box classifier or a clustering algorithm (Sec. 3.2).

significant fraction of the population on a weekly basis. While this test may not be sufficient to determine whether a particular person is contagious, the estimate of the true number of positive cases could be used by epidemiologists to monitor the reproduction number and by the health authorities to inform public policy.[2]

We advocate treating the quantification problem using Bayesian modeling, which provides uncertainty around the $P_{\text{test}}(Y)$ estimate. This uncertainty can be used directly if the distribution on the whole population is of interest, or it can be used to calibrate a probabilistic classifier to yield a more informed estimate for the label of a particular observation.

A Bayesian approach was already proposed by Storkey (2009, Sec. 6). However, that proposal relies on a generative model $P(X \mid Y)$, which can be misspecified or computationally intractable in high-dimensional settings. Hence, quantification is usually approached either via the expectation-maximization (EM) algorithm (Saerens et al., 2001) or a family of closely-related algorithms known as invariant ratio estimators (Vaz et al., 2019), black-box shift estimators (Lipton et al., 2018), or adjusted classify and count (Forman, 2008), which replace the generative model $P(X \mid Y)$ with a (potentially biased) classifier. Tasche (2017), Lipton et al. (2018), and Vaz et al. (2019) proved that these algorithms are asymptotically consistent (they rediscover $P_{\text{test}}(Y)$ in the limit of infinite data) under weak assumptions and derived asymptotic bounds on the related error.

Our contributions are:

1. We show a connection between the quantification algorithms employing a black-box (and potentially biased) classifier with Bayesian inference in a probabilistic graphical model approximating the ground-truth generative process.

2. We present a tractable Bayesian approach, which is well-suited for low-data problems. Established alternatives provide asymptotic estimates on error bounds, but may be far off for small samples (to the point that some of the estimates for $P_{\text{test}}(Y)$ may be negative). The Bayesian approach explicitly quantifies the uncertainty and does not suffer from the negative values problem. Moreover, it is possible to incorporate expert's knowledge via the choice of the prior distribution.

3. We prove that the *maximum a posteriori* inference in the considered model is asymptotically consistent under weak assumptions.

## 2 The quantification problem and existing solutions

Consider a classification problem with $\mathcal{Y} = \{1, 2, \ldots, L\}$ labels and observed features coming from a measurable space $\mathcal{X}$. A given object is then represented by two random variables (r.v.): a $\mathcal{Y}$-valued r.v. representing

---

[2]Note however that testing strategies may be adapted to the outbreaks, which in turn induce correlations between observed data, violating the usual assumption that the data are exchangeable. We discuss contraindications in Sec. 5.

the label and an $\mathcal{X}$-valued r.v. representing the measured features. We consider an anti-causal problem in which there exists a probabilistic mechanism $P(X \mid Y)$, responsible for generating the features from the label. We focus on two populations sharing the same generative mechanism, assuming that there exist probability distributions $P_{\text{train}}(X, Y)$ and $P_{\text{test}}(X, Y)$ over $\mathcal{X} \times \mathcal{Y}$ such that $P_{\text{test}}(X \mid Y = y) = P_{\text{train}}(X \mid Y = y)$ for every $y \in \mathcal{Y}$. In the literature this assumption is referred to as *prior probability shift* (Storkey, 2009). We will write $K_y^*$ for the conditional distribution $P(X \mid Y = y)$, which is the generative mechanism shared by both populations.

The *quantification problem* (González et al., 2017) asks whether given finite i.i.d. samples from the distributions $P_{\text{train}}(X, Y)$ and $P_{\text{test}}(X)$ it is possible to determine the distribution $P_{\text{test}}(Y)$. In principle, if the data are abundant, one can use samples from $P_{\text{train}}(X, Y)$ to determine the conditional distributions $K_y^*$ and then find all probability vectors $P_{\text{test}}(Y)$ which are compatible with the distribution of the features $P_{\text{test}}(X)$, written as a mixture distribution of generative mechanisms $K_y^*$:

$$P_{\text{test}}(X) = \sum_{y \in \mathcal{Y}} P_{\text{test}}(Y = y) \, K_y^*. \tag{1}$$

The uniqueness of the vector $P_{\text{test}}(Y)$ follows under strict linear independence assumption of measures $K_y^*$ (Garg et al., 2020). We review the notion of strict linear independence in Appendix A, but for a finite discrete space $\mathcal{X}$ it reduces to the linear independence of probability vectors $K_y^*$, which allows constructing the left inverse of the $P(X \mid Y)$ matrix.

In practice, however, it is not possible to fully reconstruct the distributions $P_{\text{test}}(X)$ and $K_y^*$ from finite samples and principled statistical approaches are needed. To formalize the problem, consider a probabilistic graphical model $\mathcal{M}_{\text{true}}$ in Fig. 1: probability vectors $P_{\text{train}}(Y)$ and $P_{\text{test}}(Y)$ are modeled with r.v. $\pi$ and $\pi'$ valued in the probability simplex $\Delta^{L-1} = \left\{ y \in (0, 1)^L : y_1 + \cdots + y_L = 1 \right\}$ and the ground-truth generative processes $K_y^*$ are modeled via parametric distributions $K_y(\cdot; \theta)$ with a parameter vector $\theta$. We observe $N$ pairs of r.v. $(X_i, Y_i)$ for $i \in \{1, \ldots, N\}$ sampled independently according to the model

$$Y_i \mid \pi \sim \text{Categorical}(L, \pi), \qquad X_i \mid Y_i, \theta \sim K_{Y_i}(\cdot; \theta).$$

Additionally, we observe $N'$ r.v. $X_j'$ for $j \in \{1, \ldots, N'\}$ sampled independently from the mixture distribution

$$X_j' \mid \pi', \theta \sim \sum_{y=1}^{L} \pi_y' \, K_y(\cdot; \theta),$$

or, if latent variables $Y_j'$ valued in $\mathcal{Y}$ are introduced,

$$Y_j' \mid \pi' \sim \text{Categorical}(L, \pi'), \qquad X_j' \mid Y_j', \theta \sim K_{Y_j'}(\cdot; \theta).$$

### 2.1 Likelihood-based methods

Our work draws on two major groups of quantification methods, with the description of other approaches deferred to Appendix D. The first group proceeds by considering a generative probabilistic model of the data.

Peters & Coberly (1976) assume that each $K_y(\cdot; \theta)$ is a multivariate normal distribution. Then, they estimate $\theta$ using labeled data $\{X_i, Y_i\}$ and find the maximum likelihood solution for $\pi'$ by an iterative optimization algorithm. Storkey (2009) discusses approaching quantification problems within a fully Bayesian estimation, which requires marginalization of $\theta$, and notices that such marginalization may not generally be tractable for complex generative models $K_y(\cdot; \theta)$. Moreover, a tractable generative model of high-dimensional data is likely to be misspecified, which may compromise Bayesian inference (Watson & Holmes, 2016; Lyddon et al., 2018).

Saerens et al. (2001) observe that specifying high-dimensional distributions $K_y(\cdot; \theta)$ may be avoided if one instead has access to an oracle probabilistic classifier $r \colon \mathcal{X} \to \Delta^{L-1}$ such that each $r(x) = P(Y_i \mid X_i = x, \pi)$.

Then, they show how to use a candidate value $\pi'$ to recalibrate $r(x)$ and marginalize latent variables $Y'_j$ in the Expectation-Maximization (EM) manner, which iteratively optimizes $\pi'$, targeting the maximum likelihood estimate. In Appendix D.1 we give a detailed treatment of this algorithm, together with two simple extensions: when a Dirichlet prior is used for $P(\pi')$, EM targets the *maximum a posteriori* (MAP) estimate of the posterior distribution $P(\pi' \mid \{X'_j\}, r)$. Moreover, we describe a Gibbs sampler allowing drawing from the posterior $P(\pi' \mid \{X'_j\}, r)$.

However, this posterior is generally different than $P(\pi' \mid \{X'_j\}, \{X_i, Y_i\})$, which requires marginalization over $r$, and the performance of the EM algorithm depends on the access to the oracle classifier $r$, which has to be well-calibrated (Garg et al., 2020). As modern classification methods, such as neural networks, are often miscalibrated (Guo et al., 2017), one has to leverage the labeled data set $\{X_i, Y_i\}$ to improve calibration. Alexandari et al. (2020) introduces calibration methods which can yield the state-of-the-art results using the expectation-maximization algorithm.

Although both approaches are based in sound likelihood framework, Peters & Coberly (1976) require learning high-dimensional generative models $K_y(X; \theta) = P(X \mid Y = y, \theta)$ and Saerens et al. (2001) assume an access to a well-calibrated oracle classifier $P(Y_i \mid X_i, \pi)$. Moreover, each iteration of all mentioned approaches requires operations involving all $N'$ variables $X'_j$. This may limit the scalability of either algorithm to large data sets.

## 2.2 Methods involving an auxiliary black-box classifier

The second group of approaches is based around a modification of Eq. 1 and assumes access to a given auxiliary black-box mapping: consider a given measurable space $\mathcal{C}$ and a measurable mapping[3] $f \colon \mathcal{X} \to \mathcal{C}$. For example, $f$ can be a pretrained feature extractor (such as a large language model), clustering algorithm, or a generic classifier, trained on a large data set with possibly a different set of categories.

Then, one can define new observed r.v. $C_i = f(X_i)$ and $C'_j = f(X'_j)$, which in Fig. 1 corresponds to the part of the diagram with dashed arrows. Note that the new variables act only as a summary statistic and do not increase the amount of information available. Namely, given $\{(X_i, Y_i)\}$ and $\{X'_j\}$, the r.v. $\pi'$ is independent of $\{C_i\}$ and $\{C'_j\}$, i.e.: $\pi' \perp\!\!\!\perp \{C_i\}, \{C'_j\} \,\big|\, \{(X_i, Y_i)\}, \{X'_j\}$.

However, the prior probability shift assumption implies that the distributions $P(C_i \mid Y_i = y) = P(C'_j \mid Y'_j = y)$ are equal for an arbitrary label $y \in \mathcal{Y}$ and indices $i$, $j$. In particular, Eq. 1 can be used with original features $X$ replaced with the newly introduced representations $C = f(X)$. As they are of lower dimension than $X$, it may be easier to approximate required probabilities with the available data samples.

For example, Vaz et al. (2019) propose invariant ratio estimators, which generalize earlier approaches of adjusted classify and count (Forman, 2008; Tasche, 2017) and its variant introduced by Bella et al. (2010). Namely, for a given mapping $f \colon \mathcal{X} \to \mathbb{R}^{L-1}$, one constructs

$$\hat{f}' = \frac{1}{N'} \sum_j C'_j, \quad \hat{F}_{:,y} = \frac{1}{|S_y|} \sum_{i \in S_y} C_i, \text{ where } S_y = \{i \in \{1, \ldots, N\} : Y_i = y\},$$

and solves the set of equations given by $\hat{f}' = \hat{F}\pi'$ and $\pi'_1 + \cdots + \pi'_L = 1$. In Appendix D.2 we review the closely-related algorithms employing black-box classifiers and based on matrix inversion (solving a set of linear equations), including the popular algorithm of Lipton et al. (2018).

Estimators employing black-box classifiers offer four advantages over likelihood-based methods. First, as auxiliary mapping $f$ can produce low-dimensional representations, estimating probabilities appearing in Eq. 1 may be more accurate. Secondly, Peters & Coberly (1976) require training a potentially high-dimensional generative model and Saerens et al. (2001) require a well-calibrated oracle probabilistic classifier, which may be hard to obtain in practice. Third, each optimization step in a likelihood-based method requires $O(N')$ operations. Black-box method $f$ has to be applied only once to each $X'_j$ to construct the summary statistic, which is then used for solving a linear set of equations (cf. Eq. 1). Finally, even when $P(X \mid Y)$ is not invariant

---

[3]Although for the clarity of the exposition we will use a notation corresponding to a measurable function $f$, the results hold *mutatis mutandi* for an arbitrary Markov kernel (Klenke, 2014, Sec. 8.3), so that $f$ does not need to be deterministic.

(i.e., the prior probability shift assumption does not hold), for an appropriate dimension reduction method $f$ it may hold that the distribution of low-dimensional representations, $P(C \mid Y)$, is invariant (Arjovsky et al., 2019). Lipton et al. (2018) calls invariance of $P(C \mid Y)$ the *weak prior probability shift assumption.*

However, at the same time methods employing black-box dimension reduction methods $f$ have three undesirable properties. First, dimension reduction methods may incur loss of information (Fedorov et al., 2009; Harrell, 2015, Sec. 1.3). In particular, even if the ground-truth distributions $K_y^* = P(X \mid Y = y)$ are strictly linearly independent, the pushforward distributions $P(C \mid Y = y)$ do not have to be. Secondly, solving Eq. 1 requires approximating probability distributions basing on the laws of large numbers: although these methods have desirable asymptotic behaviour, likelihood-based methods explicitly work with a given finite sample. Finally, solving a linear set of equations is not numerically stable when $P(C \mid Y)$ has large condition number. In the next section we show how to solve the last two issues within our proposed Bayesian framework.

## 3 Bayesian quantification with black-box shift estimators

We work in the setting of Fig. 1 with $N$ labeled examples $(X_1, Y_1), \ldots, (X_N, Y_N)$ and $N'$ unlabeled examples $X_1', \ldots, X_{N'}'$ obtained under the prior probability shift assumption. Additionally, we assume that we work with a given dimension reduction mapping $f \colon \mathcal{X} \to \mathcal{C}$. A fully Bayesian treatment (Storkey, 2009) relies on an assumed parametric generative mechanism $K_y(X; \theta) = P(X \mid Y = y, \theta)$ and marginalizes over all possible values of parameter $\theta$ to obtain the values of the latent variables $\pi$ and $\pi'$. From the graphical structure in Fig. 1 we note that the posterior factorizes as

$$P(\pi', \pi \mid \{(X_i, Y_i)\}, \{X_j'\}) = P(\pi \mid \{Y_i\}) \cdot P(\pi' \mid \{X_j'\}, \{(X_i, Y_i)\}),$$

and

$$P(\pi' \mid \{X_j'\}, \{(X_i, Y_i)\}) \propto P(\pi') \cdot \int \prod_i K_{Y_i}(X_i; \theta) \cdot \prod_j \sum_y \pi_y' K_y(X_j'; \theta) \, \mathrm{d}P(\theta). \tag{2}$$

The posterior $P(\pi \mid \{Y_i\})$ is analytically tractable when a Dirichlet prior $P(\pi)$ is used, so the difficulty in quantification relies in finding $P(\pi' \mid \{X_j'\}, \{(X_i, Y_i)\})$. If $\theta$ is of moderate dimension, this distribution can be approximated by using Markov chain Monte Carlo (MCMC) algorithms (Betancourt, 2017) by jointly sampling $\pi'$ and $\theta$ from the posterior $P(\pi', \theta \mid \{(X_i, Y_i)\}, \{X_j'\})$ and retaining only the $\pi'$ component. However, in complex problems involving high-dimensional $\theta$ variables and large sample sizes $N$ and $N'$, MCMC methods become computationally expensive, which may limit their applicability (Betancourt, 2015; Bardenet et al., 2017; Izmailov et al., 2021). Moreover, if parametric kernels $K_y(\cdot; \theta)$ are misspecified, which is arguably often the case in high-dimensional problems, the resulting inference may be compromised (Watson & Holmes, 2016; Lyddon et al., 2018); we investigate this issue in Sec. 4.3.

Both tractability and robustness to model misspecification can be simultaneously addressed by employing the provided black-box feature extractor $f$ to replace $X_i$ with $C_i$ and $X_j'$ with $C_j'$: Lewis et al. (2021) propose to improve robustness to model misspecification in regression models by conditioning on an insufficient summary statistic, rather than original data. In our case, we consider the conditional distribution

$$P(\pi' \mid \{C_j'\}, \{(C_i, Y_i)\}) \propto P(\pi') \cdot \int \prod_i \tilde{K}_{Y_i}(C_i; \varphi) \cdot \prod_j \sum_y \pi_y' \tilde{K}_y(C_j'; \varphi) \, \mathrm{d}P(\varphi), \tag{3}$$

where $\tilde{K}_y(\cdot; \varphi)$ are distributions on the low-dimensional space $\mathcal{C}$, rather than on high-dimensional space $\mathcal{X}$, parameterized by vector $\varphi$. Although it is possible to take $\varphi = \theta$ and define $\tilde{K}(\cdot; \varphi)$ to be the pushforward measure of $K(\cdot; \theta)$ by the dimension reduction method $f$, we generally hope that a low-dimensional distribution $\tilde{K}(\cdot; \varphi)$ may require fewer parameters and $\varphi$ will be of a much lower dimension than $\theta$, making the integral from Eq. 3 more tractable than Eq. 2.

Apart from improved tractability, conditioning on summary statistic may improve the robustness due to easier specification of low-dimensional distributions $\tilde{K}(\cdot; \varphi)$. Finally, even if the prior probability assumption does

not hold, i.e., $P(X \mid Y)$ is not invariant, the distribution of low-dimensional representations $P(C \mid Y)$ may be invariant (Arjovsky et al., 2019), which in notation of Lipton et al. (2018) corresponds to the weak prior probability shift assumption. On the other hand, conditioning on an insufficient statistic loses information: the trivial approximation $\mathcal{C} = \{1\}$ and $f(x) = 1$ forgets any available information and results in the posterior being the same as the prior, $P(\pi' \mid \{C_i, Y_i\}, \{C_j'\}) = P(\pi')$, even in the limit of infinite data.

Although the outlined methodology of approximating the intractable inference with a simpler model with a given black-box dimension reduction method $f$ is general, below we analyse the simplest possible model, where $\mathcal{C} = \{1, 2, \ldots, K\}$ and $f$ is given by a black-box classifier, or a clustering algorithm, trained on a potentially very different data set.

## 3.1 The discrete model

Consider $\mathcal{C} = \{1, 2, \ldots, K\}$ and a given black-box function $f \colon \mathcal{X} \to \mathcal{C}$. For example, $f$ can be a miscalibrated classification algorithm trained on an entirely different data set (in particular, it is possible that $K \neq L$) or a function assigning points to predefined clusters. If $K < L$, we are not able to identify $P_{\text{test}}(Y)$ basing on the outputs of $f$. However, as we find the full Bayesian posterior, rather than providing a point estimate based on matrix inversion, the posterior distribution on $\pi'$ may shrink along specific dimensions, providing accurate estimate of class prevalence for several classes. On the other hand, if $K \geq L$ and there is a strong enough correlation between outputs of $f$ and ground-truth labels, the guarantees on methods employing black-box shift classifiers and matrix inversion (Tasche, 2017; Lipton et al., 2018; Vaz et al., 2019) ensure identifiability of the prevalence vector $\pi'$ in the large data limit, as we will see in Theorem 3.1.

In this case, the model $\tilde{K}_y(\cdot; \varphi)$ is particularly simple independently of the true data-generating process $K_y^*$: as each of $\tilde{K}_y(\cdot; \varphi)$ distributions is supported on a finite set $\mathcal{C}$, they have to be categorical distributions. Namely, $\varphi = (\varphi_{yk})$ is a matrix modeling the ground-truth probability table $P(C = k \mid Y = y)$ and the model will not be misspecified provided that the weak prior probability shift assumption holds and that the prior on $\varphi$, $\pi$ and $\pi'$ is positive on the simplices $\Delta^{K-1}$ (for each $\varphi_{y:}$) and $\Delta^{L-1}$ (for $\pi$ and $\pi'$).

The approximate model $\mathcal{M}_{\text{approx}}$ takes the form

$$\pi, \pi', \varphi \sim P(\pi, \pi', \varphi)$$
$$Y_i \mid \pi \sim \text{Categorical}(L, \pi), \quad C_i \mid Y_i, \varphi \sim \text{Categorical}(K, \varphi_{Y_i:}),$$
$$Y_j' \mid \pi' \sim \text{Categorical}(L, \pi'), \quad C_j' \mid Y_j', \varphi \sim \text{Categorical}(K, \varphi_{Y_j':}).$$

We do not put specific requirements on the prior $P(\pi, \pi', \varphi)$ other than being positive on the probability simplices: for example, the Dirichlet distribution or the logistic normal distribution can be used. If precise problem-specific information is not available, we recommend using weakly informative prior distributions (Gelman et al., 2013, Sec. 2.9), such as the uniform prior over each simplex. However, we note that if the Dirichlet priors are used, the model conceptually resembles a very low-dimensional variant of latent Dirichlet allocation (Pritchard et al., 2000; Blei et al., 2003), with observed r.v. $Y_i$ and $C_i$ providing information on the $\varphi$ matrix. In Section 3.2 we will show how to construct a scalable sufficient statistic and perform efficient inference using Hamiltonian Markov chain Monte Carlo methods (Betancourt, 2017).

Although for $K < L$ the model is not identifiable, for $K \geq L$ it shares asymptotic properties similar to the alternatives based on matrix inversion. In Appendix C we prove the following result:

**Theorem 3.1.** *Assume that:*

1. *The weak prior probability shift assumption holds, i.e., $P_{train}(C \mid Y) = P_{test}(C \mid Y)$ is invariant between the populations.*

2. *The ground-truth matrix $\varphi^* = \big(P(C = k \mid Y = y)\big)_{yk}$ is of rank $L$ and all entries are strictly positive.*

3. *The ground-truth prevalence vectors $\pi^* = P_{train}(Y)$ and $\pi'^* = P_{test}(Y)$ have only strictly positive entries.*

4. *The prior $P(\pi, \pi', \varphi)$ is continuous and strictly positive on the whole space $\Delta^{L-1} \times \Delta^{L-1} \times \left(\Delta^{K-1} \times \cdots \times \Delta^{K-1}\right)$.*

*Then, for every $\delta > 0$ and $\varepsilon > 0$, there exist $N$ and $N'$ large enough such that with probability at least $1 - \delta$ the* maximum a posteriori *estimate $\hat{\pi}, \hat{\pi}', \hat{\varphi}$ is in the $\varepsilon$-neighborhood of the true parameter values $\pi^*, \pi'^*, \varphi^*$.*

Compared to the traditional approaches, we do not explicitly invert the matrix $P(C \mid Y)$ (modeled with $\varphi$), as any degeneracy is simply reflected in the posterior, showing that we did not learn anything new about the prevalence of some classes. However, if the full-rank condition holds, the *maximum a posteriori* estimate asymptotically recovers the true parameters. This result is similar to the standard results on methods employing matrix inversion, which typically assume conditions 1–3. As an additional condition, we require that the prior is positive on the whole space, which ensures that the ground-truth parameters lie in the region with positive density. This result in conceptually similar to the classical Bernstein–von Mises theorem linking Bayesian and frequentist inference in the large data limit, and in particular depends on the assumption that the model is not misspecified. However, we stress that in Bayesian statistics one should use the whole posterior distribution, rather than relying on a single point estimate. In Appendix C.1 we provide a further discussion of this point.

### 3.2 Fast inference in the discrete model

One advantage of methods employing black-box classifiers and matrix inversion over likelihood-based methods is that they require $O(N + N')$ computation time to preprocess the data, and then estimate is generated by matrix inversion, which is polynomial in $K$ and $L$. Likelihood-based methods, however, require generally at least $O(N')$ operations per each likelihood evaluation. As such, these methods may not be scalable enough to large data sets or when extensive resampling methods, such as bootstrap (Efron, 1979; Tasche, 2019), are required.

For the discrete Bayesian model, however, it is possible to preprocess the data in $O(N + N')$ time and then evaluate the likelihood and its gradient in polynomial time in terms of $K$ and $L$, without any further dependence on $N$ or $N'$. In this section we show how to construct a sufficient statistic for $\pi$, $\pi'$, and $\varphi$, whose size is independent on $N$ and $N'$ and allows one to perform efficient inference with existing sampling methods.

Define a $K$-tuple $(N'_k)_{k \in \mathcal{C}}$ of r.v. summarizing the unlabeled data set by $N'_k = \left|\{j \in \{1, \ldots, N'\} : C'_j = k\}\right|$, which can be constructed in $O(K)$ memory and $O(N')$ time. Then, for each $y \in \mathcal{Y}$, we define a $K$-tuple of r.v. $(F_{yk})_{k \in \mathcal{C}}$, such that $F_{yk} = \left|\{i \in \{1, \ldots, N\} : Y_i = y \text{ and } C_i = k\}\right|$, which requires $O(LK)$ memory and $O(N)$ time. Finally, we define an $L$-tuple of r.v. $(N_y)_{y \in \mathcal{Y}}$ by $N_y = F_{y1} + \cdots + F_{yK}$.

In Appendix B we prove that the likelihood $P(\{Y_i, C_i\}, \{C'_j\} \mid \pi, \pi', \varphi)$ is proportional to the likelihood $P\left((N_y), (N'_k), (F_{yk}) \mid \pi, \pi', \varphi\right)$ in a smaller model, $\mathcal{M}_{\text{small}}$:

$$(N_y) \mid \pi \sim \text{Multinomial}(N, \pi),$$
$$(F_{y:}) \mid N_y, \varphi \sim \text{Multinomial}(N_y, \varphi_{y:}),$$
$$(N'_k) \mid \pi', \varphi \sim \text{Multinomial}(N', \varphi^T \pi').$$

Hence, by the factorization theorem of Halmos & Savage (1949), we constructed a sufficient statistic for the inference of $\pi$, $\pi'$, $\varphi$, whose size is independent of $N$ and $N'$. In turn, we can use the likelihood of $\mathcal{M}_{\text{small}}$ (rather than $\mathcal{M}_{\text{approx}}$) to sample $\pi$, $\pi'$ and $\varphi$ from the posterior, allowing us to perform each likelihood evaluation in $O(KL)$, rather than $O(N + N')$, time. Moreover, the gradient of the likelihood is available, so we can use any of the existing Hamiltonian Markov chain Monte Carlo algorithms (Betancourt, 2017; Hoffman & Gelman, 2014).

## 4 Experimental results

We evaluate the proposed method in four aspects. In Sec. 4.1 we analyze the benefits of using Bayesian approach, rather than matrix inversion, in problems which are identifiable, but where the matrix $P(C \mid Y)$

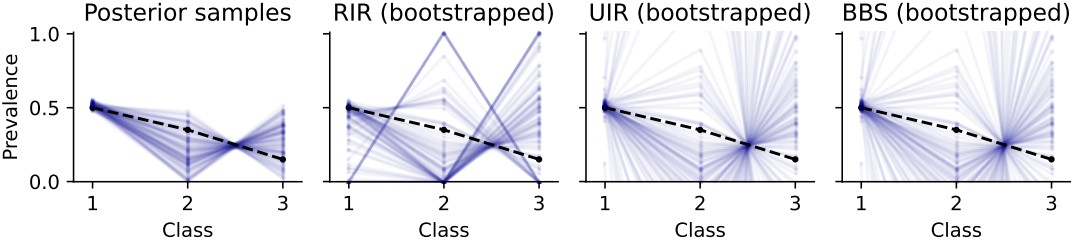

Figure 2: Uncertainty of the prevalence estimates in the nearly non-identifiable model. Samples from the proposed Bayesian posterior quantify uncertainty better than bootstrapping point estimators based on matrix inversion.

has a large condition number, requiring a large number of samples to be estimated accurately. In Sec. 4.2 we compare how the posterior mean compares with the existing point prediction methods employing black-box classifiers using extensive simulated data sets. In Sec. 4.3 we investigate how the posterior in approximate model, $P(\pi' \mid \{Y_i, C_i\}, \{C'_j\})$, compares to the posterior $P(\pi' \mid \{Y_i, X_i\}, \{X'_j\})$ with properly specified, as well as misspecified, generative models. Finally, in Sec. 4.4 we present an application of quantification methods to the problem of estimating cell type prevalence from single-cell RNA sequencing data. In all settings, we assume no specific knowledge of the problem (which could be used by principled prior elicitation) and use the uniform distributions as the priors for $\pi$, $\pi'$, and $\varphi_{y:}$ vectors.

## 4.1 Tighter estimation for hard-to-identify model

The Bayesian inference does not require an explicit inversion of the $P(C \mid Y)$ matrix. We therefore hypothesise that it may be preferable in cases where $P(C \mid Y)$ has a large condition number. Hence, in this section we consider a case with $L = K = 3$ and a given black-box classifier which can only weakly distinguish between classes 2 and 3, i.e., the ground-truth matrix $P(C \mid Y)$ is given by

$$\varphi^* = (\varphi^*_{yk}) = \begin{pmatrix} 0.96 & 0.02 & 0.02 \\ 0.02 & 0.50 & 0.48 \\ 0.02 & 0.48 & 0.50 \end{pmatrix}.$$

Although the matrix is full-rank (and asymptotic identifiability for all methods employing black-box classifiers holds), having an access to a finite sample may limit practical ability to accurately estimate the prevalence.

We simulated a data set with $N = N' = 1000$ data points and ground-truth prevalence vectors $\pi^* = (1/3, 1/3, 1/3)$ and $\pi'^* = (0.5, 0.35, 0.15)$. In Fig. 2 we plot posterior samples from the proposed Bayesian model together with the bootstrapped (Efron, 1979) predictions of three methods employing black-box classifiers and performing explicit inversion: restricted and unrestricted invariant ratio estimators (RIR and UIR respectively; Vaz et al. (2019)) and black-box shift estimator of Lipton et al. (2018) (BBS). We used $S = 100$ bootstrap samples using the stratified bootstrapping procedure introduced for quantification problems by Tasche (2019); in Appendix E we additionally reproduce this experiment varying the sampled data sets.

We see that the Bayesian approach identifies the component $\pi'_1$, leaving large uncertainty on entries $\pi'_2$ and $\pi'_3$. On the other hand, all bootstrapped methods struggle with estimating any of the components. Moreover, UIR and BBS often result in estimates with negative entries. As we further illustrate in Appendix E, this behaviour is typical for low sample sizes, and bootstrapped predictions become stable for $N = N' = 10^4$ samples. However, the proposed Bayesian approach does not suffer from these low-data issues, appropriately quantifying uncertainty.

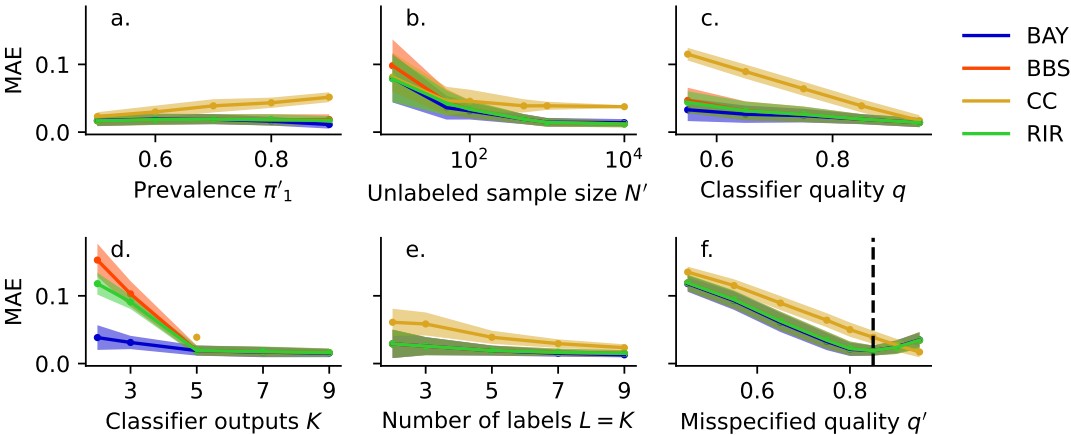

Figure 3: Mean absolute error for quantification using simulated categorical black-box classifiers under different scenarios, the proposed Bayesian approach (BAY) shown in blue.

## 4.2 Simulations employing the discrete model

Although we advocate for quantifying uncertainty around the prediction of $\pi'$, quantification is typically posed as a point estimation problem. We therefore compare the point predictions of three matrix-inversion methods mentioned before (RIR, UIR, and BBS) with the posterior mean in the Bayesian model (BAY) and a simple baseline known as "classify and count" (CC; Tasche (2017) proves that this method may not converge to the ground-truth value even in the high-data limit).

**Experimental design** In this paragraph we describe the experimental design with the default parameter values. They are changed one at a time, as described in further sections, and for each setting we simulate labeled and unlabeled data sets $S = 50$ times and, for each method, we record the mean absolute error (MAE) between the ground-truth value $\pi'^{*}$ and the point estimate $\hat{\pi}'$. Using root mean squared error (RMSE) does not qualitatively change the results (see Appendix E.2).

We fix the data set sizes $N = 10^{3}$ and $N' = 500$ and use $L = K = 5$ as a default setting. The ground-truth prevalence vectors are parametrized as $\pi^{*} = (1/L, \ldots, 1/L)$ and $\pi'^{*}(r) = \left(r, \frac{1-r}{L-1}, \ldots, \frac{1-r}{L-1}\right)$. By default, we use $r = 0.7$. The ground-truth matrix $P(C \mid Y)$ is parameterized as $\varphi_{yy}^{*} = q$ and $\varphi_{yk}^{*} = (1-q)/(K-1)$ for $k \neq y$ and $K \geq L$, with the default value $q = 0.85$. Whenever $K < L$ we use $\varphi_{yk}^{*} = 1/K$ for $y \in \{L+1, L+2, \ldots, K\}$ to obtain a valid probability vector.

**Changing prevalence** We investigate the impact of increasing the prior probability shift (the difference between $\pi$ and $\pi'$) by changing $r = \pi_{1}' \in \{0.5, 0.6, 0.7, 0.8, 0.9\}$ and summarize the results in Fig. 3a. CC is adversely impacted by a strong data shift. The other estimators all perform similar to each other.

**Changing data set size** We investigate whether the algorithms converge to the ground-truth value in the large data limit. We vary $N' \in \{10, 50, 100, 500, 10^{3}, 10^{4}\}$. As shown in Fig. 3b, the large data limit appears very similar (except for CC), agreeing with asymptotic identifiability guarantees for BBS, RIR and our MAP estimates, although BBS appears slightly less accurate than the others in a low data regime.

**Changing classifier quality** We investigate the impact of classifier quality by changing it in range $q \in \{0.55, 0.65, 0.75, 0.85, 0.95\}$ and show the results in Fig. 3c. All considered method converge to zero error for high quality, but the convergence of CC is much slower than for the other algorithms.

**Changing the classification granularity** We change $K \in \{2, 3, 5, 7, 9\}$, creating a setting when a given black-box classifier, trained on a different data distribution, is still informative about some of the classes, but provides different information. In particular, the CC estimator cannot be used for $K \neq L$. Although the original formulation of BBSE and IR assumes $K = L$, we proceed with least square error solution. Our choice

Figure 4: First two plots: conditional Student distributions $P(X \mid Y)$ and the training and test populations. Third plot: posterior samples under four different Bayesian models. Fourth plot: coverage of credible intervals.

of $\varphi^*$ given above guarantees that the classifier for $K > L = 5$ will contain at least as much information as a classifier with a smaller number of classes. Conversely for $K < L$, the information about some of the classes will be insufficient even in the large data regime — it is not possible for the matrix $P(C \mid Y)$ to have rank $L$, and asymptotic consistency does not generally hold.

The results are shown in Fig. 3d. While all methods considered (apart from CC) suffer little error for $K \geq L$, we note that our model-based approach can still learn something about the classes for which the classifier is informative enough, while the techniques based on matrix inversion are less effective. Additionally, we should stress that the Bayesian approach gives the whole posterior distribution on $\pi'$ (which can still appropriately shrink along well-recoverable dimensions).

**Changing the number of classes** Finally, we jointly change $L = K \in \{2, 3, 5, 7, 9\}$. We plot the results in Fig. 3e. Again, classify and count obtains markedly worse results, with smaller differences between the other methods.

**Model misspecification** Finally, we study whether the considered approaches are robust to breaking the weak prior probability shift assumption: the unlabeled data are sampled according to a different $P(C \mid Y)$ distribution, parameterized by $q'$. The weak prior probability shift assumption corresponds to the setting $q' = q$, which is marked with a dashed black line in Fig. 3f. Although in this case asymptotic identifiability guarantees do not hold, we believe this to be an important case which may occur in practice (when additional distributional shifts are present).

We see that the performance of BBS, IR and BAY estimates deteriorates for large discrepancies between $q$ and $q'$. However, for $|q - q'| \leq 0.05$, the median error of BBS, IR and BAY is still arguably tame, so we hope that these methods can be employed even if the prior probability shift assumption is only approximately correct. Note that in the case when $q' > q$, (i.e., the classifier has better predictive accuracy on the unlabeled data set than on the labeled data set, which we think rarely occurs in practice), CC outperforms other methods.

### 4.3 Uncertainty assessment in a misspecified model of a mixture of Student distributions

As mentioned in Sec. 3.1, using a black-box function $f \colon \mathcal{X} \to \mathcal{C}$ to reduce the dimensionality to a set $\{1, 2, \ldots, K\}$ not only improves the tractability of the problem, but also has the potential to make the model more robust to misspecification by replacing the prior probability shift assumption (invariance of $P(X \mid Y)$ with its weak version (invariance of $P(C \mid Y)$) and by learning the parameters of a categorical discrete distribution, rather than parameters of a potentially misspecified distribution $K_y(X; \theta)$. On the other hand, $\mathcal{M}_{\text{approx}}$ loses information from the problem, so we do not expect it to be as appropriate as a properly specified generative model for $P(X \mid Y)$.

To investigate properies of the $\mathcal{M}_{\text{approx}}$ approach we generated low-dimensional data, so that Bayesian inference in $\mathcal{M}_{\text{true}}$ is still tractable: we consider a mixture of two Student t-distributions presented in Fig. 4 from which we sample $N = N' = 10^3$ points. We implemented a Gaussian mixture model and a Student mixture distribution in NumPyro (Phan et al., 2019), setting weakly-informative priors on their parameters (see Appendix E.3). For the $\mathcal{M}_{\text{approx}}$ we partitioned the real axis into $K$ bins: $(-\infty, -4), [-4, a_1), [a_1, a_2), \ldots, [a_{K-3}, 4), [4, \infty)$, with all intervals (apart from the first and the last one) of equal length.

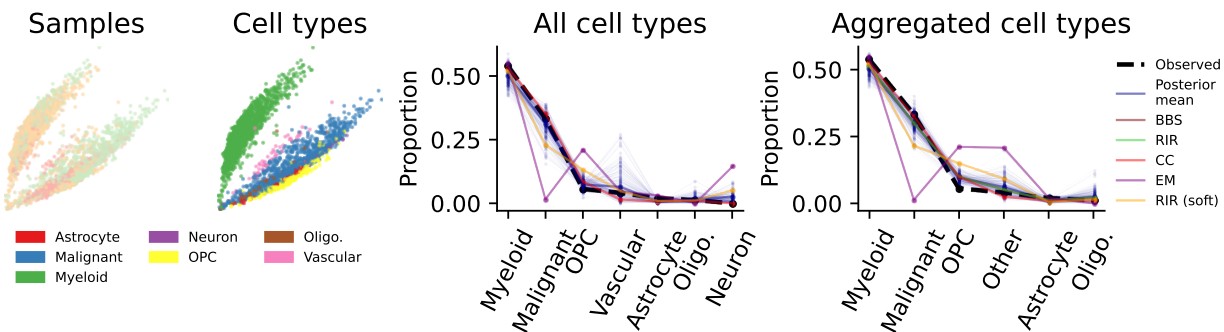

Figure 5: First two panels: principal components of the feature vectors $X$ colored by the biopsy sample and cell types. Third panel: inferred cell type proportions. Fourth panel: inferred cell type proportions with vascular cells and neurons merged into one type.

In Fig. 4 we see that using 10 bins yields very similar posterior to the one of a well-specified model and that using 5 bins yields a wider posterior, which agrees with the perspective that discretization resulted in information loss. However, a misspecified Gaussian mixture model concentrates around a wrong value.

We repeated the simulation $S = 200$ times, excluded the runs with convergence issues (see Appendix E.3), and checked the frequentist coverage of the highest-density credible intervals. The coverage of the discretized models as well as of the properly specified Student mixture agrees well with the expected value. However, the credible intervals from the misspecified Gaussian mixture are systematically too narrow. As we demonstrate in Appendix E.3, misspecification is less problematic for lower sample sizes and more problematic for large sample sizes.

We conclude that, in the considered setting, conditioning on an insufficient statistic, as proposed by Lewis et al. (2021) for regression problems, is a viable solution also for quantification. However, it is not as data-efficient as a properly-specified generative model if one is available. We did not compare obtained posteriors in high-dimensional problems due to high computational costs associated with running MCMC on high-dimensional generative models.

## 4.4 Prevalence estimation in real world data

As a more practical application of the proposed quantification method, we consider single-cell RNA sequencing data. Darmanis et al. (2017) collected biopsy specimens from four glioblastoma multiforme tumors corresponding to four different populations of cells. Each cell belongs to one of six healthy types (astrocyte, neuron, oligodendrocyte, OPC, myeloid or vascular) or is a malignant cancer cell, yielding in total $L = 7$ distinct cell types.

Each cell is sequenced to yield a gene expression vector $X$ with 23,368 entries. Basing on gene expression vectors and known marker genes, the cells have been assigned to the considered cell types we treat provided annotations as ground-truth labels. In Fig. 5 we plotted the first two principal components of the whole data set to visualise the distribution of features within each sample and with relation to the cell type. We note that although Darmanis et al. (2017) used TPM normalization (Zhao et al., 2021) to normalize the features $X$ and the cells seem to roughly cluster within cell types, the sample-specific effects are still visible (see also Appendix E.4), so that the prior probability shift assumption, of invariant $P(X \mid Y)$, is violated.

We consider a semi-realistic scenario in which one wants to estimate cell prevalence in an automated fashion employing a given black-box cell type classifier. We treat the first two samples as an auxiliary cell atlas on which a generic black-box cell type classifier was trained (we use a random forest), the third sample as an available labeled data set, $\{(X_i, Y_i)\}$, and the fourth sample as an unlabeled data set, $\{X'_j\}$, for the quantification problem. Although the prior probability shift is violated, we can hope that the labels predicted by the random forest are more invariant and that methods working under the weak prior probability shift assumption may still yield reasonable estimates.

We consider quantification method using predicted labels (posterior mean in the proposed model, BBS, RIR, and CC) as well as two methods accepting probabilities, rather than labels: Expectation-Maximization (EM) and a soft variant of the restricted invariant ratio estimator (RIR (soft)). We do not use recalibration techniques proposed by Alexandari et al. (2020), using the vanilla probabilities provided by the random forest.

Generally, the posterior mean captures well the true cell types prevalence, showing large uncertainty around the vascular cells and neurons. Similar performance is obtained by the simple CC baseline, owing to the good performance of the classifier. Interestingly, the methods using estimated probabilities (RIR (soft) and EM) obtained the worst performance. We hypothesise that this is due to the violated prior probability assumption and that using discrete labels, rather than continuous probabilities, improves the robustness.

As methods employing matrix inversion and a black-box classifier (BBS and RIR) failed due to non-invertibility of the estimated matrices, we then decided to merge two least prevalent cell types occurring in the data set (vascular cells and neurons) into a single "Other" class. In that case, RIR and BBS obtain performance on par with posterior mean and CC.

## 5 Discussion

The presented approach generalizes point estimates provided by black-box shift estimators and invariant ratio estimators to the Bayesian inference setting. This allows one to *quantify uncertainty* and *use existing knowledge* about the problem by prior specification. Moreover, by the construction of the sufficient statistic our approach is tractable even in large-data limit (for either data set considered). In all our experiments, the suggested estimator obtained at least as good performance as the existing methods, outperforming them in the $K < L$ case where the number of modeled classes differs from the "true" number of classes. Compared to point estimates with asymptotic guarantees, our approach "knows what it does not know", meaning that the posterior is meaningful even if the matrix $P(C \mid Y)$ is not (left-)invertible, and it is specific for the prevalence values of those classes for which the feature extractor $f$ is sufficiently informative.

Moreover, the proposed approach aligns with the approaches employing black-box feature extractors $f$, which can be trained on a different data set and may be not calibrated properly. This is particularly useful when a hard, fully black-box classifier is given without the possibility of retraining it, which is an increasingly common theme with modern AI applications, which are often huge assets doing sophisticated processing, and also often proprietary and only available through APIs.

However, the method we introduce is not free from challenges. As in all Bayesian inferences, care is required regarding modeling assumptions: whether the discrete model is applicable and what prior should be used. In particular, even the weak prior probability shift assumption may not hold (e.g., if the labeled and unlabeled data sets were collected under radically different conditions or the labeled and unlabeled data sets have different classes $\mathcal{Y}$). Additionally, Bayesian inference often carries a model choice problem, and different choices for $K$ or the discretization method $f$ may yield different posteriors on the prevalence vector $\pi'$, especially in the low data regime. If the generative model $P(X \mid Y, \theta)$ is well-specified and tractable, we suggest to use this instead of an approximation $P(C \mid Y, \varphi)$. If it is not tractable, we suggest to use the available classifier with $K$ classes, observing the quality of $\varphi = P(C \mid Y)$ matrix, and perhaps training one's own classifier on some hold-out data set. The experiments performed in Section 4.2 and Theorem 3 of Lipton et al. (2018) suggest that improving the classification accuracy results in more accurate prevalence quantification. Finally, we rely on the Markov chain Monte Carlo sampling schemes, which explore an $O(KL)$-dimensional parameter space. Although in this manuscript we focus on problems with a moderate number of classes, there exist complex data sets, such as ImageNet (Deng et al., 2009), with thousands of categories. We note that this can pose additional computational challenges for the MCMC samplers (Betancourt, 2015; Bardenet et al., 2017; Izmailov et al., 2021), which we do not investigate in this work.

**Statement of broader impact**

This article discusses a Bayesian method of quantifying the prevalence of different classes in an unlabeled data set. We note that in general the parameter posterior conditioned on the full data view $X$ can be different from the posterior conditioned on some representation $C = f(X)$ — in cases where a reliable model $P(X \mid Y)$ is available and the inference is tractable, we suggest to use this instead of our discretized method. Secondly, the model need not apply — perhaps prior probability shift is not the only distribution shift occurring in the problem or the data may not be exchangeable. In epidemiology, for example, outbreaks induce correlations between the healthiness of different people that can easily extend to sampling. Finally, even if all the assumptions hold, recalibrating a probabilistic classifier with quantification may have undesirable consequences regarding fairness (Plecko & Bareinboim, 2022).

**Code availability and reproducibility**

We ensured reproducibility by designing all experiments as Snakemake workflows (Mölder et al., 2021). The code and workflows used to run the experiments and generate the figures are available in the `https://github.com/pawel-czyz/labelshift` repository.

**Acknowledgments**

We would like to thank Ian Wright for valuable comments on the manuscript. This publication was supported by GitHub, Inc. and ETH AI Center. We would like to thank both institutions.

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

# A  Strict linear independence of measures

We use the following definition:

**Definition A.1.** *Let $K_1, \ldots, K_L$ be probability measures on a measurable space $\mathcal{X}$. We say that they are strictly linearly independent* if for every non-zero $\lambda \in \mathbb{R}^L$, there exists a measurable set $\mathcal{A}_\lambda$ such that

$$\lambda_1 K_1(A_\lambda) + \cdots + \lambda_L K_L(A_\lambda) \neq 0.$$

This definition is especially useful for proving identifiability of a well-specified mixture model:

**Theorem A.2.** *Let $K_1, \ldots, K_L$ be strictly linearly independent probability measures on space $\mathcal{X}$.*

*Then, for every two vectors of mixture weights $\pi, \pi' \in \Delta^{L-1}$ such that*

$$\pi_1 K_1 + \cdots + \pi_L K_L = \pi_1' K_1 + \cdots + \pi_L' K_L$$

*it follows that $\pi = \pi'$.*

*Proof.* Consider the difference $\lambda = \pi - \pi' \in \mathbb{R}^L$. If it was not the zero vector, then for some set $A_\lambda$ we would have

$$\pi_1 K_1(A_\lambda) + \cdots + \pi_L K_L(A_\lambda) \neq \pi_1' K_1(A_\lambda) + \cdots + \pi_L' K_L(A_\lambda).$$

Note also that $A_\lambda$ is not of positive measure with respect to both of the mixture measures, meaning that for a well-specified model the discrepancy will eventually be visible according to the law of large numbers. $\square$

Although the definition above looks different from the original one proposed by Garg et al. (2020), they are essentially equivalent:

**Lemma A.3.** *Let $\mu$ be a $\sigma$-finite measure on $\mathcal{X}$ such that the Radon–Nikodym derivatives $k_y = \mathrm{d}K_y/\mathrm{d}\mu$ exist. Then, the probability measures $K_1, \ldots, K_L$ are strictly linearly independent if and only if*

$$\int_{\mathcal{X}} \left| \sum_{y \in \mathcal{Y}} k_y(x) \right| \mathrm{d}\mu(x) \neq 0$$

*for every non-zero $\lambda \in \mathbb{R}^L$.*

*Proof.* Consider any $\lambda \neq 0$ and write $\nu = \lambda_1 K_1 + \cdots + \lambda_K K_L$ for the signed measure. Using the standard rules of Radon–Nikodym calculus, the condition on the integral is equivalent to $|\nu|(\mathcal{X}) \neq 0$. If there exists $A_\lambda$ such that $\nu(A_\lambda) \neq 0$, then $|\nu|(\mathcal{X}) \geq |\nu|(A_\lambda) \geq |\nu(A_\lambda)| > 0$.

Conversely, assume that $|\nu|(\mathcal{X}) \neq 0$ and take the Hahn decomposition of $\mathcal{X}$, with $\mathcal{X} = P \cup N$ such that $P \cap N = \varnothing$, $\nu(P) \geq 0$, and $\nu(N) \leq 0$. We define now Hahn–Jordan decomposition of $\nu$ into two positive measures, $\nu^+(A) = \nu(A \cap P)$ and $\nu^-(A) = -\nu(A \cap N)$, with the properties $\nu = \nu^+ - \nu^-$ and $|\nu| = \nu^+ + \nu^-$. We conclude that $|\nu|(\mathcal{X}) = \nu(P) - \nu(N) \neq 0$, so that at least one of the sets $P$ and $N$ can be taken as $A_\lambda$. $\square$

The above characterization of strict linear independence gives the following result:

**Lemma A.4.** *Assume that $\mathcal{X}$ is a standard Borel space and $\mu$ is a strictly positive measure. Further, assume that the Radon–Nikodym derivatives $k_1, \ldots, k_L$ are continuous functions, treated as vectors in the space of all continuous functions $C(\mathcal{X}, \mathbb{R})$. Then, if $k_1, \ldots, k_L$ are linearly independent, then $K_1, \ldots, K_L$ are strictly linearly independent.*

*Proof.* Take any $\lambda \neq 0$ and write $u = |\lambda_1 k_1 + \cdots + \lambda_L k_L| \in C(\mathcal{X}, \mathbb{R})$. From the linear independence it follows that there exists $x_0 \in \mathcal{X}$ such that $u(x_0) > 0$.

We can use the continuity of $u$ to find an open neighborhood $A$ of $x_0$ such that for all $x \in A$ we have $u(x) > u(x_0)/2$. As $u$ is non-negative and $\mu$ is strictly positive, we have $\mu(A) > 0$ and

$$\int_X u(x)\,\mathrm{d}\mu(x) \geq \int_A u(x)\,\mathrm{d}\mu(x) \geq \frac{u(x_0)}{2} \cdot \mu(A) > 0.$$

$\square$

## B   Derivation of the sufficient statistic

Starting from the joint probability

$$P(\pi, \pi', \varphi, \{Y_i, C_i\}, \{Y_j', C_j'\}) = P(\pi, \pi', \varphi) \times \prod_{i=1}^{N} P(C_i \mid \varphi, Y_i) P(Y_i \mid \pi) \times \prod_{j=1}^{N'} P(C_j' \mid \varphi, Y_j') P(Y_j' \mid \pi'),$$

we need to derive

$$P(\pi, \pi', \varphi \mid \{Y_i, C_i\}, \{C_j'\}) \propto P(\{Y_i, C_i\}, \{C_j'\} \mid \pi, \pi', \varphi) P(\pi, \pi', \varphi),$$

The observed likelihood is given by marginalization of $Y_j'$ variables:

$$P(\{Y_i, C_i\}, \{C_j'\} \mid \pi, \pi', \varphi) = \sum_{l_{N'} \in \mathcal{Y}} \cdots \sum_{l_1 \in \mathcal{Y}} \prod_{i=1}^{N} P(C_i \mid \varphi, Y_i) P(Y_i \mid \pi) \prod_{j=1}^{N'} P(C_j' \mid \varphi, Y_j' = l_j) P(Y_j' = l_j \mid \pi)$$

$$= \underbrace{\prod_{i=1}^{N} P(C_i \mid \varphi, Y_i) P(Y_i \mid \pi)}_{A} \times \underbrace{\left( \sum_{l_{N'} \in \mathcal{Y}} \cdots \sum_{l_1 \in \mathcal{Y}} \prod_{j=1}^{N'} P(C_j' \mid \varphi, Y_j' = l_j) P(Y_j' = l_j \mid \pi') \right)}_{B}.$$

Each of these terms will be calculated separately.

We want to calculate

$$A := \prod_{i=1}^{N} P(C_i = c_i \mid \varphi, Y_i = y_i) P(Y_i = y_i \mid \pi) = \underbrace{\prod_{i=1}^{N} P(C_i = c_i \mid \varphi, Y_i = y_i)}_{A_1} \times \underbrace{\prod_{i=1}^{N} P(Y_i = y_i \mid \pi)}_{A_2}.$$

The term $A_2$ is simple to calculate: as $P(Y_i = y_i \mid \pi) = \pi_{y_i}$, we have

$$A_2 = \prod_{i=1}^{N} \pi_{y_i} = \prod_{l=1}^{L} (\pi_l)^{n_l},$$

where $n_l$ is the number of $i \in \{1, \ldots, N\}$, such that $y_i = l$. In particular, up to a factor $N!/n_1! \ldots n_L!$, this is the PMF of the multinomial distribution parametrised by $\pi$ evaluated at $(n_1, \ldots, n_L)$.

To calculate $A_1$ we need to observe that $P(C_i = k \mid \varphi, Y_i = l) = \varphi_{lk}$. Hence,

$$A_1 = \prod_{i=1}^{N} P(C_i = c_i \mid \varphi, Y_i = y_i) = \prod_{l=1}^{L} \prod_{k=1}^{K} (\varphi_{lk})^{f_{lk}},$$

where $f_{lk}$ is the number of $i \in \{1, \ldots, N\}$, such that $y_i = l$ and $c_i = k$. Observe that $n_l = f_{l1} + \cdots + f_{lK}$.

In particular, up to the factor

$$\prod_{l=1}^{L} \frac{n_l!}{f_{l1}! \ldots f_{lK}!}$$

this corresponds to the product of PMFs of $L$ multinomial distributions parametrised by probabilities $\varphi_{l:}$ evaluated at $f_{l:}$.

Recall that

$$B := \sum_{l_{N'} \in \mathcal{Y}} \cdots \sum_{l_1 \in \mathcal{Y}} \prod_{j=1}^{N'} P(C'_j = c'_j \mid \varphi, Y'_j = l_j) P(Y'_j = l_j \mid \pi').$$

We can use the sum-product identity

$$\sum_{l_{N'} \in \mathcal{Y}} \cdots \sum_{l_1 \in \mathcal{Y}} \prod_{j=1}^{N'} f_j(l_j) = \prod_{j=1}^{N'} \sum_{l \in \mathcal{Y}} f_j(l)$$

to reduce:

$$B = \prod_{j=1}^{N'} \sum_{l \in \mathcal{Y}} P(C'_j = c'_j \mid \varphi, Y'_j = l) P(Y'_j = l \mid \pi').$$

Because both $C'_j$ and $Y'_j$ are parametrised with categorical distributions, we have

$$P(C'_j = k \mid \varphi, Y'_j = l) = \varphi_{lk}$$

and

$$P(Y'_j = l \mid \pi') = \pi'_l,$$

so

$$\sum_{l \in \mathcal{Y}} P(C'_j = k \mid \varphi, Y'_j = l) P(Y'_j = l \mid \pi') = (\varphi^T \pi')_k.$$

Hence,

$$B = \prod_{j=1}^{N'} (\varphi^T \pi')_{c'_j} = \prod_{k=1}^{K} \left( (\varphi^T \pi')_k \right)^{n'_k},$$

where $n'_k$ is the number of $j \in \{1, \ldots, N'\}$ such that $c'_j = k$. In particular, up to a factor of $N'!/n'_1! \cdots n'_K!$, this is the PMF of the multinomial distribution parametrized by probabilities $\varphi^T \pi'$ evaluated at $(n'_1, \ldots, n'_K)$.

## C   Proof of asymptotic identifiability

In this section we prove Theorem 3.1. We first need to establish two simple lemmas regarding approximate left inverses:

**Lemma C.1.** *Choose any norms on the space of linear maps $\mathbb{R}^L \to \mathbb{R}^K$ and $\mathbb{R}^K \to \mathbb{R}^L$. Suppose $K \geq L$ and that $A_0 \colon \mathbb{R}^L \to \mathbb{R}^K$ is of full rank $L$. Then, for every $\varepsilon > 0$ there exists $\delta > 0$ such that if $A \colon \mathbb{R}^L \to \mathbb{R}^K$ is any matrix such that*

$$||A - A_0|| < \delta,$$

*then the left inverse $A^{-1} := (A^T A)^{-1} A^T$ exists and*

$$||A^{-1} - A_0^{-1}|| < \varepsilon.$$

*Proof.* First note that indeed the choice of norms does not matter, as all norms on finite-dimensional vector spaces are equivalent.

Then, observe that rank is a lower semi-continuous function, so that for sufficiently small $\delta$ the map $A$ will be of rank $L$ as well.

Finally, it is clear that the chosen formula for the left inverse is continuous as a function of $A$.   $\square$

**Lemma C.2.** *If $K \geq L$ and matrix $A_0 \colon \mathbb{R}^L \to \mathbb{R}^K$ is of full rank $L$, then for every $\varepsilon > 0$ there exist numbers $\delta > 0$ and $\nu > 0$ such that for every linear mapping $A \colon \mathbb{R}^L \to \mathbb{R}^K$ and vector $v \in \mathbb{R}^L$ if*

$$||A - A_0|| < \delta$$

*and*

$$||Av - A_0 v_0|| < \nu,$$

*then*

$$||v - v_0|| < \varepsilon.$$

*Proof.* Again, the norm on either space can be chosen arbitrarily without any loss of generality. We will choose the $p$-norm for vectors and the induced matrix norms.

From the previous lemma we know that for any chosen $\beta > 0$ we can take $\delta > 0$ such that $A$ is left-invertible and

$$||B - B_0|| < \beta,$$

where $B = A^{-1}$ and $B_0 = A_0^{-1}$ are the left inverses in the form defined before.

Write $w = Av$ and $w_0 = A_0 v_0$. We have

$$
\begin{aligned}
||v - v_0|| &= ||Bw - B_0 w_0|| \\
&= ||(Bw - B_0 w) + (B_0 w - B_0 w_0)|| \\
&= ||(B - B_0)w + B_0(w - w_0)|| \\
&\leq ||(B - B_0)w|| + ||B_0(w - w_0)|| \\
&\leq ||B - B_0|| \cdot ||w|| + ||B_0|| \cdot ||w - w_0|| \\
&\leq \beta ||w|| + ||B_0|| \nu.
\end{aligned}
$$

We can bound each of these two terms by $\varepsilon/3$ choosing appropriate $\beta$ and $\nu$. Then, we can find $\delta$ yielding appropriate $\beta$. $\square$

Now the proof of Theorem 3.1 will proceed in two steps:

1. We show than for any prescribed probability we can find $N$ and $N'$ large enough that the maximum likelihood solution will be close to the true parameter values.

2. Then, we show that for reasonable priors the maximum a posteriori solution will almost surely assymptotically converge to the maximum likelihood solution.

Let's assume that the data was sampled from the model with true parameters $\pi^*$, $\pi'^*$, $\varphi^*$ and take $\delta > 0$ and $\varepsilon > 0$.

For any $\nu > 0$ we can use the fact that log-likelihood is given by

$$\ell(\pi, \pi', \varphi) = \sum_{l \in \mathcal{Y}} N_l \log \pi_l + \sum_{k \in \mathcal{C}} \sum_{l \in \mathcal{Y}} F_{lk} \log \varphi_{lk} + \sum_{k \in \mathcal{C}} N'_k \log(\varphi^T \pi')_k,$$

and by the strong law of large numbers we can find $N$ and $N'$ large enough that with probability at least $1 - \delta$ we will have $||\hat{\pi} - \pi^*|| < \nu$ and $||\hat{\varphi} - \varphi^*|| < \nu$, and $||\hat{\varphi}^T \hat{\pi}' - \varphi^{*T} \pi'^*|| < \nu$, where $\hat{\pi}$, $\hat{\varphi}$, and $\hat{\pi}'$ is the maximum likelihood estimate.

Basing on the previously established lemmas we conclude that we can pick $\nu$ small enough that $||\hat{\pi} - \pi^*|| < \varepsilon$, $||\hat{\varphi} - \varphi^*|| < \varepsilon$, and $||\hat{\pi}' - \pi'^*|| < \varepsilon$.

Now note that if we assume the PDF of the prior $P(\pi, \pi', \varphi)$ to be continuous, we can take a compact neighborhood of $(\pi^*, \pi'^*, \varphi^*)$ inside $\Delta^{L-1} \times \Delta^{L-1} \times \Delta^{K-1} \times \cdots \times \Delta^{K-1}$ with probability mass arbitrarily close to 1. Then, the log-prior defined on this set will be bounded and the *maximum a posteriori* estimate can be made arbitrarily close to the maximum likelihood estimate with any desired probability.

### C.1 Should one rely on the *maximum a posteriori* estimate?

Although in the above section we discuss why the *maximum a posteriori* (MAP) estimate is consistent in the large-sample limit, we advise against using it: as the data is finite, the fully Bayesian approach is to use the full posterior (approximated by the Markov chain Monte Carlo samples) to understand the associated uncertainty on the provided estimates.

In settings in which the posterior distribution has to be summarized in terms of a single point estimate (e.g., for the comparison with estimators providing point estimates in Section 4.2), we advise for using the posterior mean, which is available directly from the MCMC samples. Conversely, the MAP estimate requires optimization, rather than sampling, and may not be well-defined (e.g., in non-identifiable problems with non-invertible $P(C \mid Y)$ matrix or when the sample size is not large enough).

Moreover, posterior mean may be preferable over MAP on the basis of Bayesian decision theory. We present a short version of the standard decision-theoretic argument and refer to Bernardo & Smith (1994, Sec. 5.1.5) for a more detailed treatment.

An approach employed in Bayesian decision theory is to define a loss function $\ell(\hat{\pi}', \pi'^*)$ quantifying risk associated with choosing the estimate $\hat{\pi}'$ when in reality the estimand attains $\pi'^*$ value. Various applications involve different loss functions, to be specified by the decision maker: for example, if the component $\pi'_1$ describes the prevalence of a viral disease, one may prefer to choose a loss function assigning higher loss to the predictions underestimating the disease prevalence (i.e., $\hat{\pi}'_1 < \pi'^*_1$ should result in a higher loss than $\hat{\pi}'_1 > \pi'^*_1$). The Bayesian decision theory argues then to provide an estimate minimizing the expected loss, $\int \ell(\hat{\pi}', \pi') \, \mathrm{d}P(\pi' \mid \text{data})$.

The MAP estimate (provided that it exists) corresponds to the limit of the losses $\ell_{0-1,\epsilon}(\hat{\pi}', \pi') = \mathbf{1}[||\hat{\pi}' - \pi'|| > \epsilon]$ for $\epsilon \to 0^+$. The resulting discontinuous $0-1$ loss may not be of direct interest in the applied problems: for finite data the estimates are expected to be different from the ground-truth value and one instead may try to quantify by how much amount the estimate differs. For an $\ell_2(\hat{\pi}', \pi') = ||\hat{\pi}' - \pi'||^2$ loss, the minimum of the expected loss is attained at the posterior mean.

At the same time, we stress that it is preferable to analyze the whole posterior, rather than to rely on a single point estimate, especially that in reality the decision maker may not have an access to an oracle loss function and can consider different candidate losses. In particular, in Section 4.2 and Appendix E.2 we quantify the algorithm performance using two losses for which the minimum may not be attained at the posterior mean. Nevertheless, the experimental results presented there confirm that the (arguably suboptimal) choice of the posterior mean as the point estimate is on par with the point estimates provided by other methods.

## D  Quantification algorithms

In this section we provide additional details on existing quantification methods, expanding on the description in Sec. 2.

### D.1 Expectation-maximization and the Gibbs sampler

In this section we analyse the expectation-maximization algorithm introduced by Saerens et al. (2001), which assumes access to a well-calibrated probabilistic classifier providing the probabilities $r(x) = P(Y \mid X = x, \pi)$. We assume a Dirichlet prior for $P(\pi')$, so that the expectation-maximization targets maximum a posteriori of the distribution $P(\pi' \mid \{X'_j\}, r)$. The original algorithm of Saerens et al. (2001) corresponds then to the uniform prior, $P(\pi') = \text{Dirichlet}(\pi' \mid 1, 1, \dots, 1)$. Finally, we will show how to adjust the expectation-maximization algorithm to obtain a Gibbs sampler, sampling from the posterior $P(\pi' \mid \{X'_j\}, r)$. To improve readability we will generally drop conditioning on $r$, leaving it implicit.

### D.1.1 Expectation-maximization

Saerens et al. (2001) notice that if one has a well-calibrated classifier $P(Y \mid X, \pi)$, then they also have an access to a distribution $P(Y \mid X, \pi')$:

$$
\begin{aligned}
P(Y = y \mid X = x, \pi') &\propto P(Y = y, X = x \mid \pi') \\
&= P(X = x \mid Y = y, \pi') P(Y = y \mid \pi') \\
&= P(X = x \mid Y = y) \, \pi'_y,
\end{aligned}
$$

where the proportionality constant does not depend on $y$. Analogously,

$$
P(Y = y \mid X = x, \pi) \propto P(X = x \mid Y = y) \, \pi_y.
$$

As $P(X = x \mid Y = y)$ is the same, we can take the ratio of both expressions and obtain

$$
P(Y = y \mid X = x, \pi') \propto P(Y = y \mid X = x, \pi) \frac{\pi'_y}{\pi_y},
$$

where the proportionality constant does not depend on $y$. This yields unnormalized probabilities, which can be easily rescaled so that they sum up to 1.

Expectation-maximization is an iterative algorithm finding a stationary point of the log-posterior

$$
\begin{aligned}
\log P(\pi' \mid \{X'_j = x'_j\}) &= \log P(\pi') + \log P(\{X'_j = x'_j\} \mid \pi') \\
&= \log P(\pi') + \sum_{j=1}^{N'} \log P(X'_j = x'_j \mid \pi').
\end{aligned}
$$

In particular, by running the optimization procedure several times, we can aim at finding the maximum a posteriori estimate. Assume that at the current iteration the proportion vector is $\pi'^{(t)}$. Then,

$$
\begin{aligned}
\log P(X'_j = x'_j \mid \pi') &= \log \sum_{y=1}^{L} P(X'_j = x'_j, Y'_j = y \mid \pi') \\
&= \log \sum_{y=1}^{L} P(Y'_j = y \mid \pi^{(t)}, X'_j = x'_j) \frac{P(X'_j = x'_j, Y'_j = y \mid \pi')}{P(Y'_j = y \mid \pi'^{(t)}, X'_j = x'_j)} \\
&\geq \sum_{y=1}^{L} P(Y'_j = y \mid \pi'^{(t)}, X'_j = x'_j) \log \frac{P(X'_j = x'_j, Y'_j = y \mid \pi')}{P(Y'_j = y \mid \pi'^{(t)}, X'_j = x'_j)}
\end{aligned}
$$

where the bound follows from Jensen's inequality. Hence,

$$
\begin{aligned}
\log P(\{X'_j = x'_j\} \mid \pi') &= \sum_{j=1}^{N'} \log P(X'_j = x'_j \mid \pi') \\
&\geq \sum_{j=1}^{N'} \sum_{y=1}^{L} P(Y'_j = y \mid \pi'^{(t)}, X'_j = x'_j) \log \frac{P(X'_j = x'_j, Y'_j = y \mid \pi')}{P(Y'_j = y \mid \pi'^{(t)}, X'_j = x'_j)}.
\end{aligned}
$$

Now let

$$
Q(\pi, \pi^{(t)}) = \log P(\pi) + \sum_{j=1}^{N'} \sum_{y=1}^{L} P(Y'_j = y \mid \pi^{(t)}, X'_j = x'_j) \log \frac{P(X'_j = x'_j, Y'_j = y \mid \pi)}{P(Y'_j = y \mid \pi^{(t)}, X'_j = x'_j)},
$$

be a lower bound on the log-posterior. We will define the value $\pi'^{(t+1)}$ by optimizing this lower bound, i.e., $\pi'^{(t+1)} := \mathrm{argmax}_{\pi'} Q(\pi', \pi'^{(t)})$.

Define auxiliary variables $\xi_{jy} = P(Y'_j = y \mid \pi'^{(t)}, X'_j = x'_j)$, which can be calculated using the probabilistic classifier $r$ as above. Hence,

$$Q(\pi', \pi'^{(t)}) = \log P(\pi') + \sum_{j=1}^{N'} \sum_{y=1}^{L} \left( \xi_{jy} \log P(X'_j = x'_j, Y'_j = y) - \xi_{jy} \log \xi_{jy} \right)$$

Note that the term $\xi_{jy} \log \xi_{jy}$ does not depend on $\pi'$, so it does not have to be included in the optimization. Similarly, we can write $\log P(X'_j = x'_j, Y'_j = y \mid \pi') = \log P(X'_j = x'_j \mid Y'_j = y) + \log \pi'_y$ and notice that $\log P(X'_j = x'_j \mid Y'_j = y)$ also does not depend on $\pi'$. Hence, we have to optimize the expression

$$\log P(\pi') + \sum_{j=1}^{N'} \sum_{y=1}^{L} \xi_{jy} \log \pi'_y,$$

where $P(\pi')$ is modeled as the Dirichlet distribution, $\text{Dirichlet}(\pi \mid \alpha_1, \ldots, \alpha_L)$. Hence, the optimization objective becomes

$$\sum_{y=1}^{L} \left( (\alpha_y - 1) + \sum_{j=1}^{N'} \xi_{jy} \right) \log \pi'_y,$$

with the constraint $\pi'_1 + \cdots + \pi'_L = 1$. Saerens et al. (2001) use the technique of Lagrange multipliers. However, we can optimise the first $L-1$ coordinates and write $\pi'_L = 1 - (\pi'_1 + \cdots + \pi'_{L-1})$. In this case, if we differentiate with respect to $\pi'_y$, we obtain $A_y/\pi'_y - A_L/\pi'_L = 0$, where $A_y = \alpha_y - 1 + \sum_{j=1}^{N'} \xi_{jy}$.

Hence, $\pi'_y = k A_y$ for some constant $k > 0$. As

$$\sum_{y=1}^{L} A_y = \sum_{y=1}^{L} \alpha_y - L + \sum_{j=1}^{N'} \sum_{y=1}^{L} \xi_{jy} = \sum_{y=1}^{L} \alpha_y - L + N',$$

we obtain

$$\pi'_y = \frac{1}{(\alpha_1 + \cdots + \alpha_L) + N' - L} \left( \alpha_y - 1 + \sum_{j=1}^{N'} \xi_{jy} \right),$$

which is taken as the next iteration value, $\pi'^{(t+1)}$. The procedure is repeated until the sequence $\pi'^{(t)}$ (approximately) converges to a point.

### D.1.2 Gibbs sampler

As typical for expectation-maximization algorithms, it is possible to implement a Gibbs sampler targeting the sample from the posterior $P(\pi' \mid \{X'_j\})$, rather than the mode (maximum a posteriori).

The Gibbs sampler will iteratively sample from the high-dimensional $P(\pi', \{Y'_j\} \mid \{X'_j\})$ distribution. Note that for a Dirichlet prior we have

$$P(\pi' \mid \{Y'_j = y_j, X'_j\}) = \text{Dirichlet} \left( \pi' \middle| \alpha_1 + \sum_{j=1}^{N'} \mathbf{1}[y_j = 1], \ldots, \alpha_L + \sum_{j=1}^{N'} \mathbf{1}[y_i = L] \right).$$

The assignments of individual points are then sequentially sampled as

$$Y'_k \sim P(Y'_k \mid \{Y'_1, \ldots, Y'_{k-1}, Y'_{k+1}, \ldots, Y'_L\}, \{X'_j\}, \pi'),$$

which is possible due to the equality

$$P(Y'_k \mid X'_k, \pi) = \text{Categorical}(\xi_{k1}, \ldots, \xi_{kL}),$$

where $\xi_{ky} = P(Y'_k = y \mid X'_k = x_k, \pi')$, which is obtained using the probabilistic classifier $r$ similarly as above.

### D.2    Estimators employing auxiliary black-box classifiers

In this section we briefly review the main algorithms employing a black-box classifier.

#### D.2.1    Classify and count

When $\mathcal{C} = \mathcal{Y}$ and $f\colon \mathcal{X} \to \mathcal{Y}$ is a classifier trained for a given problem with good accuracy, the simplest approach is to count its predictions and normalize by the total number of examples in the unlabeled data set. However, as Tasche (2017) shows, this approach does not need to correctly estimate $P(Y)$ even in the limit of infinite data.

#### D.2.2    Adjusted classify and count

Consider a case of an imperfect binary classifier, with $\mathcal{Y} = \mathcal{C} = \{+, -\}$. The true and false positive rates are defined by

$$\mathrm{TPR} = P(C = + \mid Y = +)$$
$$\mathrm{FPR} = P(C = + \mid Y = -)$$

and can be estimated using the labeled data set.

If $\theta = P_{\mathrm{test}}(Y = +)$, we have

$$P_{\mathrm{test}}(C = +) = \mathrm{TPR} \cdot \theta + \mathrm{FPR} \cdot (1 - \theta)$$

which can be estimated by applying the classifier to the unlabeled data set and counting positive outputs.

If we assume that $\mathrm{TPR} \neq \mathrm{FPR}$, i.e., the classifier has any predictive power, we obtain

$$\theta = \frac{P_{\mathrm{test}}(C = +) - \mathrm{FPR}}{\mathrm{TPR} - \mathrm{FPR}}.$$

Then, $P_{\mathrm{test}}(C = +)$ is estimated by counting the predictions of the classifier on the unlabeled data set. As Tasche (2017) showed, it is consistent in the limit of infinite data.

Two generalizations, extending it to the problems with more classes, are known as the invariant ratio estimator and black-box shift estimator.

#### D.2.3    Invariant ratio estimator

Vaz et al. (2019) introduce the invariant ratio estimator, generalizing the Adjusted Classify and Count approach as well as the "soft" version of it proposed by Bella et al. (2010).

Consider any function $f\colon \mathcal{X} \to \mathbb{R}^{L-1}$. For example, if $u\colon \mathcal{X} \to \mathcal{Y}$ is a classifier predicting outputs in the set $\{1, \ldots, L\}$, we may define $f$ as the one-hot encoding of $L - 1$ labels and assign the zero vector to the last label:

$$f(x) = \begin{cases} (1, 0, \ldots, 0) & \text{if } u(x) = 1, \\ (0, 1, \ldots, 0) & \text{if } u(x) = 2, \\ \quad \vdots \\ (0, 0, \ldots, 1) & \text{if } u(x) = L - 1, \\ (0, 0, \ldots, 0) & \text{if } u(x) = L. \end{cases}$$

Analogously, for a soft classifier $u\colon \mathcal{X} \to \Delta^{L-1} \subset \mathbb{R}^L$, $f$ may be defined as $f_k(x) = u_k(x)$ for $k \in \{1, \ldots, L - 1\}$.

Then the *unrestricted* estimator $\hat{\pi}' \in \mathbb{R}^L$ is given by solving the linear system

$$
\begin{cases}
\hat{F}_{11}\pi'_1 + \cdots + \hat{F}_{1L}\pi'_L & = \hat{f}'_1 \\
\qquad\qquad \vdots \\
\hat{F}_{L-1,1}\pi'_1 + \cdots + \hat{F}_{L-1,L}\pi'_L & = \hat{f}'_{L-1} \\
\pi'_1 + \cdots + \pi'_L & = 1
\end{cases}
$$

where

$$
\hat{f}'_k = \frac{1}{N'} \sum_{j=1}^{N'} g_k(x'_j)
$$

and

$$
\hat{F}_{kl} = \frac{1}{|S_l|} \sum_{x \in S_l} g_k(x),
$$

where $S_l$ is the subset of the labeled data set with $y_i = l$.

Note that adjusted classify and count is a special case of the invariant ratio estimator, for a hard classifier. Similarly, the algorithm proposed by Bella et al. (2010) is a special case of invariant ratio estimator for a soft classifier.

The generalization for $K \neq L$ is immediate, with $\hat{G}$ becoming a $(K-1) \times L$ matrix and $\hat{g}$ becoming a vector of dimension $K - 1$. Finally, Vaz et al. (2019) introduce a restricted estimator $\hat{\pi}'_R \in \Delta^{L-1}$, which is given by a projection of $\hat{\pi}'_U$ onto the probability simplex. In our implementation we use the projection via sorting algorithm (Shalev-Shwartz & Singer, 2006; Blondel et al., 2014).

### D.2.4 Black-box shift estimator

Black-Box shift estimators are also based on the observation that

$$
P_{\text{test}}(C) = P(C \mid Y)P_{\text{test}}(Y),
$$

where $P(C \mid Y)$ matrix can be estimated using either labeled or the unlabeled data set. Instead of solving this matrix equation directly by finding the (left) inverse, Lipton et al. (2018) estimate the pointwise ratio $R(Y) = P_{\text{test}}(Y)/P_{\text{train}}(Y)$ by rewriting this equation as

$$
P_{\text{test}}(C) = P_{\text{train}}(C, Y)R(Y),
$$

and estimate the joint probability matrix $P_{\text{train}}(C, Y)$ using the labeled data set. Then, the equation can be solved for $R(Y)$. By pointwise multiplication by $P_{\text{train}}(Y)$ (estimated using the labeled data set) the prevalence vector $P_{\text{test}}(Y)$ is found.

Note that this approach naturally generalizes to the $K \neq L$ case. Lipton et al. (2018) study the case $K = L$ and derive asymptotic error bounds. More rencently, Azizzadenesheli et al. (2019) introduced a regularized variant of this approach.

### D.2.5 Unsupervised recalibration

Ziegler & Czyż (2020) study the quantification problem from the perspective of recalibration of a given probabilistic classifier. Their method can be interpreted as partly a black-box shift estimator and partly as a likelihood-based estimator. Namely, they propose to use a black-box classifier to predict the labels $C_i = f(X_i)$ and $C'_j = f(X'_j)$ and estimate the probability table $P(C \mid Y)$ by using the plug-in estimator. However, they note that solving explicitly Eq. 1 may suffer from numerical issues when condition number is high and instead they optimize the multinomial likelihood on the observed counts $C'_j$.

### D.3 Other algorithms

Other quantification methods include the CDE-Iterate algorithm of Xue & Weiss (2009), which can obtain good empirical performance on selected problems (Karpov et al., 2016). However, as Tasche (2017, Sec. 3.4) showed, it is not asymptotically consistent. Zhang et al. (2013) describes a kernel mean matching approach, with a provable theoretical guarantee. However, as Lipton et al. (2018, Sec. 6) observed, kernel-based methods may be challenging to scale to large data sets. Finally, Moreo et al. (2021) present a Python package for quantification problems.

## E Experimental details and additional experiments

In this section we provide additional details on experimental protocols used in Sec. 4 together with additional experimental results. Experiments described in Appendices E.1, E.4, and E.5 were run on a laptop with 32 GiB RAM and 16 CPU cores clocked at 4680 MHz and finished under six hours. Experiments described in Appendices E.2 and E.3 require runs over many random seeds and require larger computing power (unless the number of random seeds is reduced). We ran them sequentially on a cluster equipped with 384 GiB RAM and 128 CPU cores clocked at 2.25–3.7 GHz. As the cluster is shared between different researchers and uses the Slurm workload manager (Yoo et al., 2003) to distribute runs among different projects, this provides an upper bound on the computational resources used. These experiments have finished in five hours.

### E.1 Nearly non-identifiable model

We reproduced the experiment described in Sec. 4.1 $S = 5$ times varying the random seed to obtain different data samples (and, subsequently, different posterior and bootstrap samples) as well as the data set size under $N = N'$ constraint. We noted that methods based on matrix inversion raised an error whenever a matrix estimated from the bootstrap sample was singular. We dropped such bootstrap samples.

In Fig. 6 we use $N = N' = 100$, in Fig. 7 we use $N = N' = 10^3$, and in Fig. 8 we use $N = N' = 10^4$.

We generally see that bootstrap for $N = N' \in \{10^2, 10^3\}$ can result in negative prevalence estimates. On the other hand, restricted invariant ratio estimator (RIR) often does not appropriately estimate the first component. Hence, we consider the Bayesian posterior preferable in low-data settings. For $N = N' = 10^4$ we see that the performance of all methods is comparable.

For each simulated data set, we ran four Markov chains with 500 warm-up steps and 1000 samples each using the NUTS algorithm of Hoffman & Gelman (2014). To flag potential convergence issues, we calculated the potential scale reduction factor $\hat{R}$ (Gelman et al., 2013, Sec. 11.4). For each variable it did not exceed 1.01. The effective sample size was over 3,000 for all parameters and across all experimental settings.

### E.2 Discrete categorical model benchmark

The default parameters introduced in Sec. 4.2 have been gathered in Table 1. For each data set we used four Markov chains with 500 warm-up steps and 1000 samples collected per chain. The maximal $\hat{R}$ did not exceed 1.01 and the minimal effective sample size was 772.

Fig. 9 represents the outcomes of the experiment where mean absolute error metric has been replaced with the root mean squared error:

$$\text{RMSE}(\hat{\pi}', \pi'^*) = \sqrt{\frac{1}{L} \sum_y (\hat{\pi}'_y - \pi'^*_y)^2}.$$

Qualitatively, the conclusions do not change.

### E.3 Misspecified model

We repeated the experiment described in Sec. 4.3 for $N = N' \in \{10^2, 10^3, 10^4\}$ samples with the results presented in Fig. 10. For each generated data set and each method we ran four Markov chains with 1,500

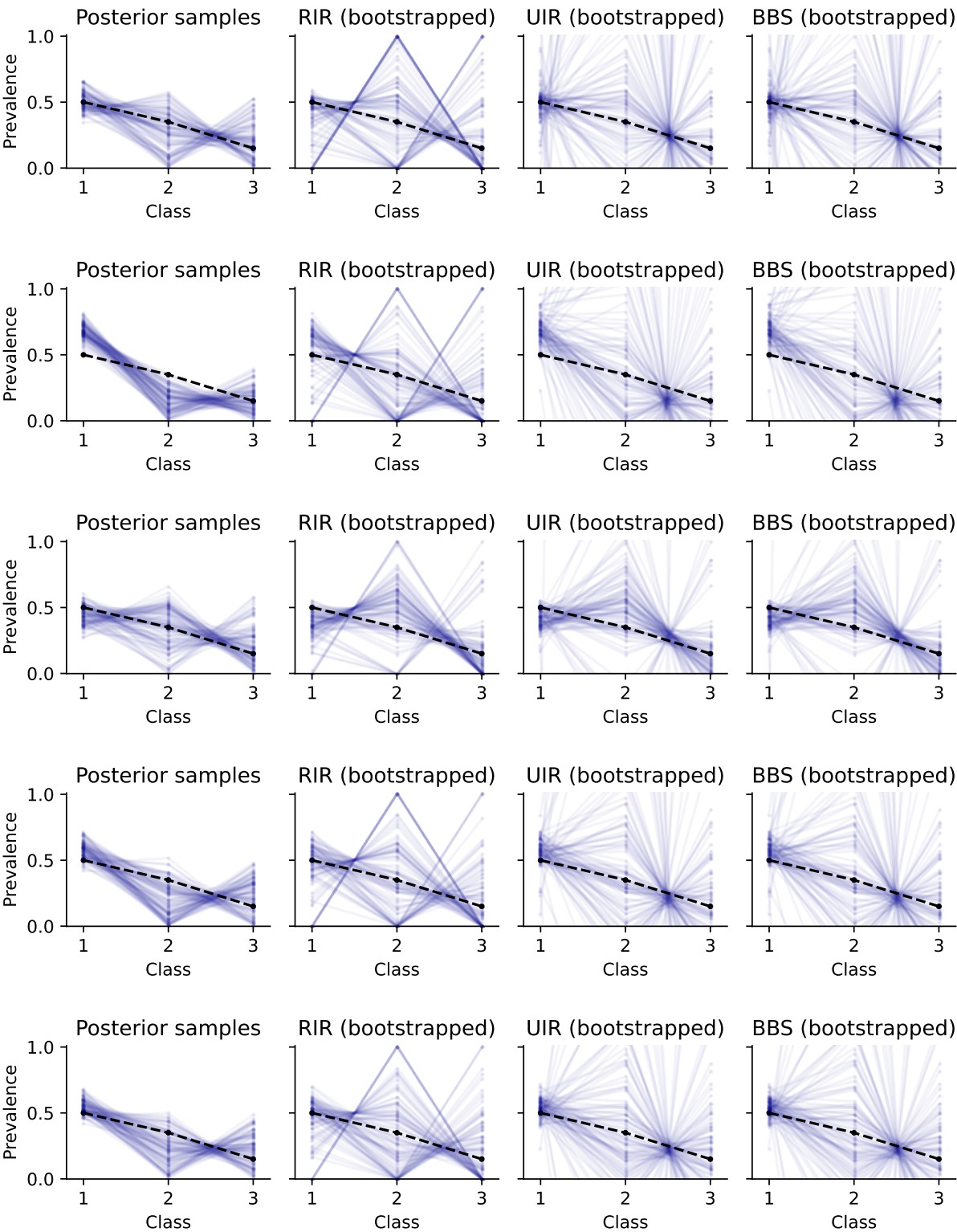

Figure 6: For $N = N' = 100$ samples the posterior is not very precise around the first component. We note that for unrestricted estimators the bootstrap samples result in negative probability estimates.

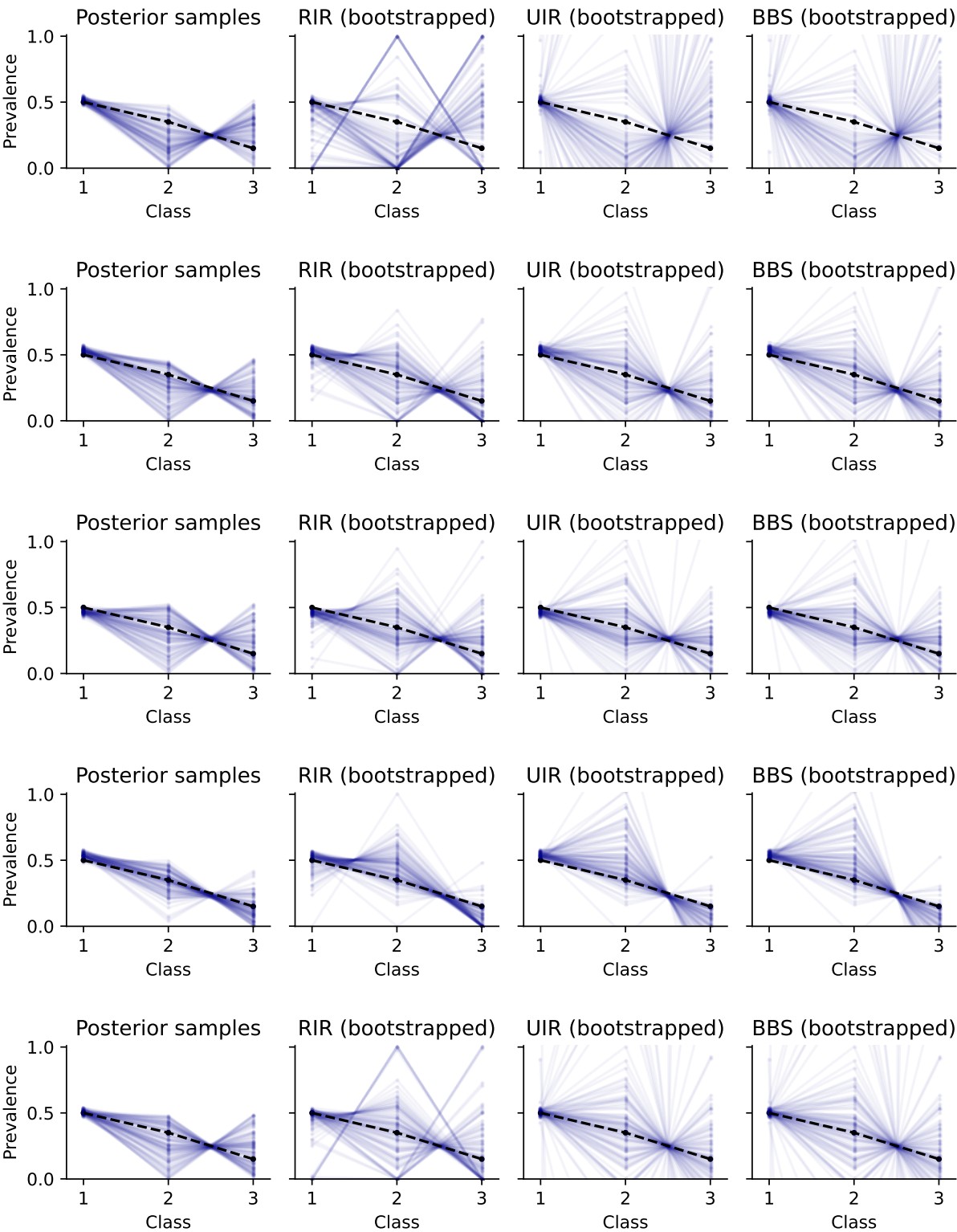

Figure 7: For $N = N' = 10^3$ Bayesian posterior concentrates around the ground-truth value of the first component. However, bootstrap samples often yield negative probability estimates.

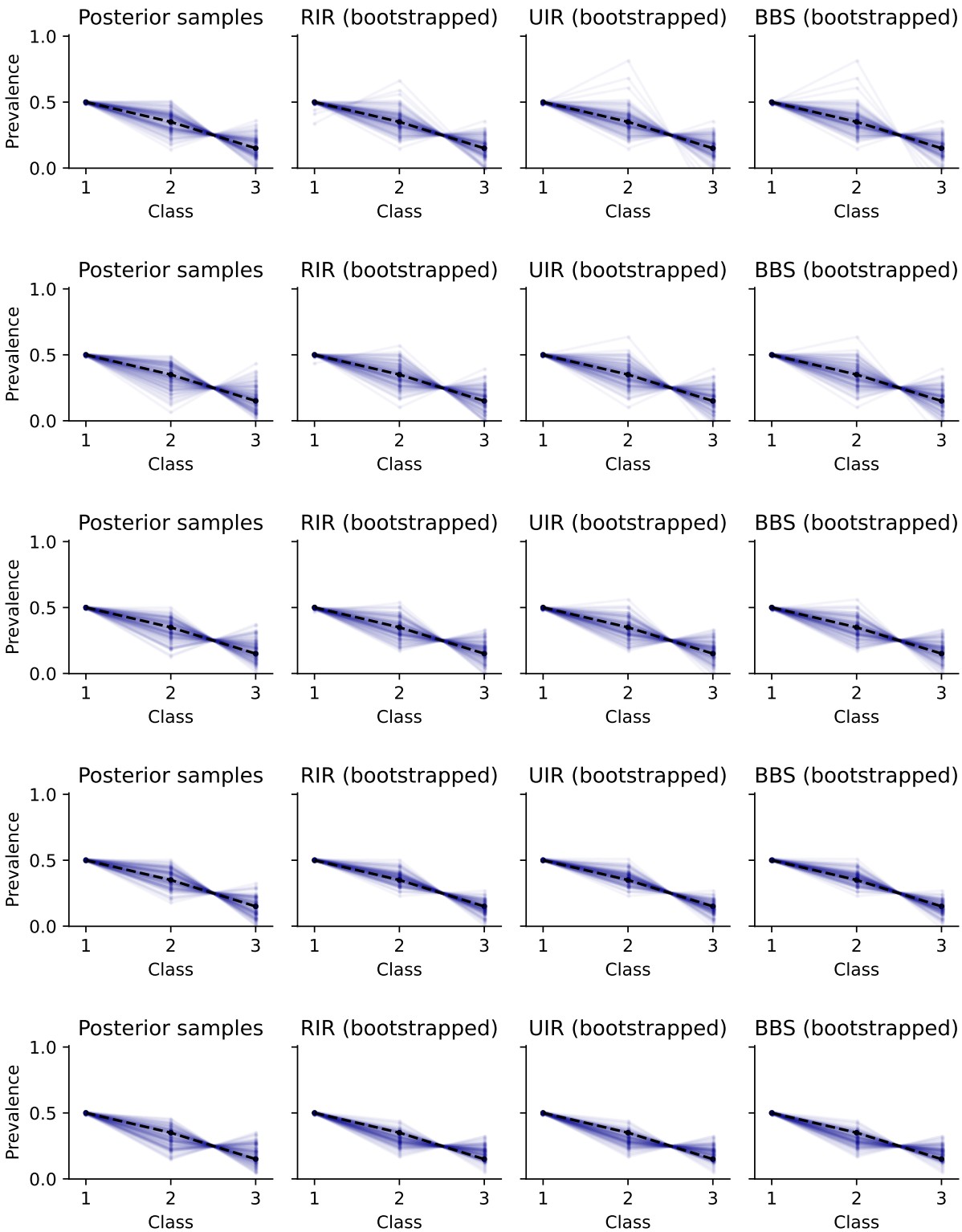

Figure 8: For $N = N' = 10^4$ samples the first component is perfectly determined. Bootstrap samples capture the uncertainty well and are qualitatively similar to the samples from the Bayesian posterior.

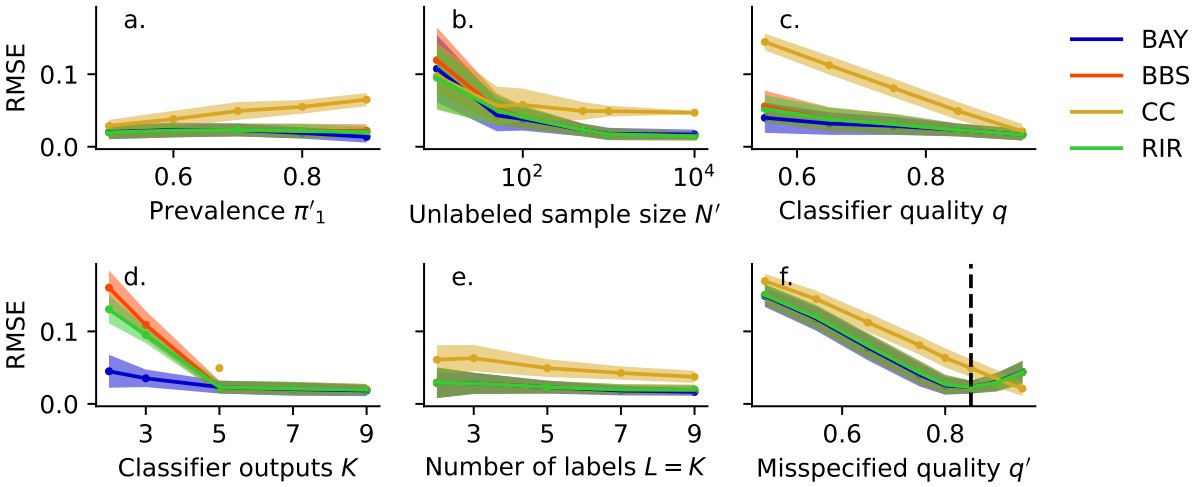

Figure 9: Quantification using simulated categorical black-box classifiers under different scenarios.

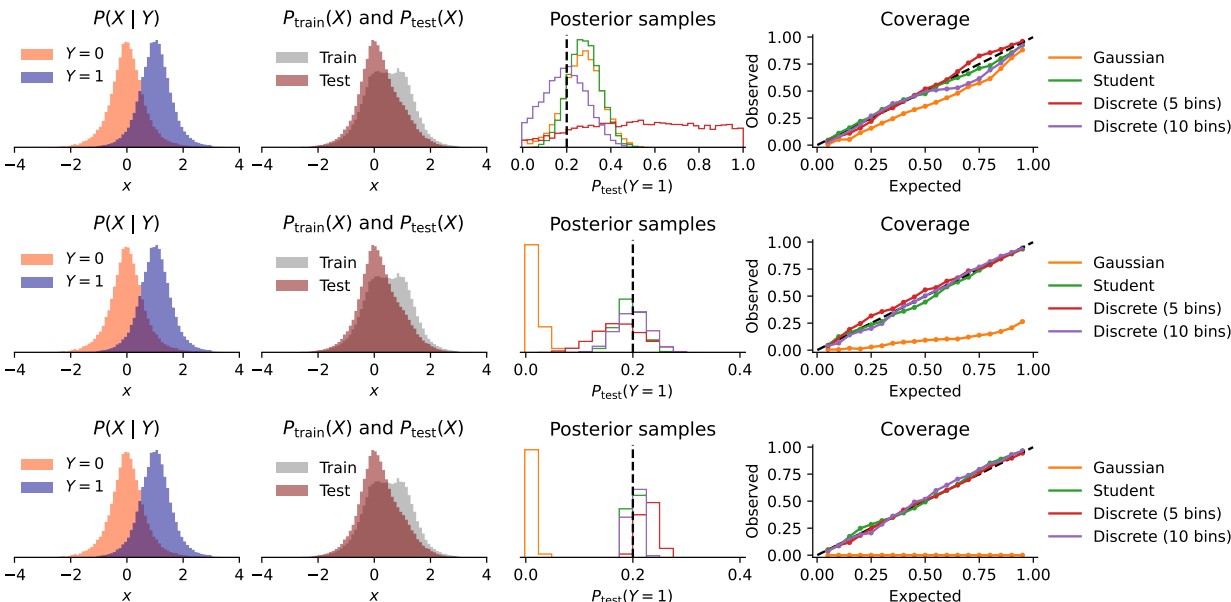

Figure 10: Experiments with a mixture of Student distributions. First column: conditional Student distributions $P(X \mid Y)$. Second column: train (labeled) and test (unlabeled) distributions. Third column: posterior according to differend models. Fourth column: coverage of high-density credible intervals measured over $S = 200$ simulations. Top row: $N = N' = 100$ samples. Middle row: $N = N' = 10^3$ samples. Bottom row: $N = N' = 10^4$ samples.

warm-up steps and 2,000 samples collected per chain. We noticed occasional convergence issues: runs with at least one parameter such that $\hat{R} \geq 1.02$ were excluded from the coverage calculation (see Table 2). The convergence for all models is satisfactory (with the minimal number of simulations retained being 157 out of conducted 200), although we noticed that more runs were excluded for larger sample sizes and that misspecification (i.e., the Gaussian mixture model) resulted in more runs to be excluded.

We see that the coverage of the high-density credible intervals of the discrete model generally agrees with the nominal value. However, the posterior in the discrete model can be wider than in the properly specified model using the generative mechanisms $P(X \mid Y)$. Moreover, we see that using discretized quantification method may be preferable over using a misspecified model when the large sample size is used: although for $N = N' = 100$ the (misspecified) Gaussian mixture model has coverage close to the nominal value, for $N = N' \in \{10^3, 10^4\}$ the coverage deteriorates quickly.

**Parameters of the ground-truth mixture model**  We used $(X \mid Y = 1) \sim \mathcal{T}(0, 0.5^2, 3)$ and $(X \mid Y = 2) \sim \mathcal{T}(1, 0.5^2, 4)$, where $\mathcal{T}(\mu, \sigma^2, \nu)$ is the location-scale $t$-distribution, i.e., the pushforward distribution of the standard Student $t$-distribution $\mathcal{T}(0, 1, \nu)$ with $\nu$ degrees of freedom by the affine mapping $x \mapsto \mu + \sigma x$.

We sampled the number of labeled samples with label $Y = 1$ using $N_1 \sim \mathrm{Binomial}(N, 0.5)$ and then defined the number of samples with label $Y = 2$ by $N - N_1$. Similarly, we generated an unlabeled data set, but with the class prevalence 0.2, rather than 0.5.

**Priors on the Gaussian and Student mixture models**  In both cases we used the uniform prior, $\pi' \sim \mathrm{Dirichlet}(1, 1)$, on the prevalence vector. We modeled the scale parameters $\sigma_i \sim |\mathcal{C}|(1)$ via the half-Cauchy prior and the location parameters via $\mu_i \sim \mathcal{N}(0, 3^2)$. Additionally, the Student mixture had a positive prior on the degrees of freedom, $\nu_i \sim \Gamma(1, 1)$. The Gaussian mixture model can be treated as a special case of this model with the constraint $\nu_i = \infty$ for both components.

### E.4    Single-cell data analysis

We downloaded the TPM-normalized (Zhao et al., 2021) data sequenced by Darmanis et al. (2017) from the Curated Cancer Cell Atlas. We applied the $x \mapsto \log(1 + x)$ transform to all entries.

In Fig. 11 we visualize $P(X \mid Y)$ by projecting the gene expression $X$ on the first four principal components (calculated using all samples pooled together). We see that the distribution $P(X \mid Y)$ differs between the samples, although the cell types seem to roughly cluster together and the random forest classifier may distinguish well between different subtypes.

As a random forest we used the SciKit-Learn implementation (Pedregosa et al., 2011, v. 1.4.1) with default hyperparameters and 20 trees. Before training the random forest we reduced the dimensionality by projecting the training data onto the first 50 principal components. This projection (onto the components defined by the training data) is used for making the predictions on other samples, before a random forest is applied.

Similarly as in App. E.1, in this experiment we ran four chains with 500 warm-up steps and 1000 samples each. To flag potential convergence issues, we calculated the potential scale reduction factors $\hat{R}$ to two decimal places, which did not exceed 1.02. The minimal corresponding effective sample size was 298.

### E.5    Sensitivity to the prior choice

In Sec. 4 we used uniform priors over all simplices, which is supposed to act as a reference prior (Gelman et al., 2013, Sec. 2.8) for problems without precise domain-specific information. In practice, the inference

Table 1: Default parameters used in the experiments.

| $N$ | $N'$ | $r$ | $q$ | $L$ | $K$ |
|-----|------|-----|------|-----|-----|
| 1000 | 500 | 0.7 | 0.85 | 5 | 5 |

Table 2: Number of simulations (out of 200) with $\hat{R} < 1.02$ for each sample size $N = N'$ and model.

| Sample Size | Discrete (10 bins) | Discrete (5 bins) | Gaussian | Student |
|---|---|---|---|---|
| 100 | 200 | 200 | 192 | 200 |
| 1000 | 199 | 199 | 174 | 200 |
| 10000 | 199 | 196 | 157 | 197 |

Figure 11: Projections onto first four principal components of the whole data set. Each column describes the projections of a specified sample on the 1st and 2nd (first row) or the 3rd and 4th (second row) component, coloured by the cell type. We see distributional differences and conclude that $P(X \mid Y)$ is not invariant between samples.

may depend on the choice of the prior and a sensitivity analysis should be performed (see Kruschke (2021) or Gelman et al. (2020, Sec. 6.3)).

We illustrate the prior choice problem in an example with $L = K = 2$ classes with an imperfect classifier $P(C = 1 \mid Y = 1) = P(C = 2 \mid Y = 2) = 0.7$ and prevalence vectors $\pi = (0.5, 0.5)$ and $\pi' = (0.7, 0.3)$. We sampled three data sets, differing by the number of points, $N = N' \in \{50, 500, 5000\}$ and we fitted three models, differing by the choice of the prior.

We used the symmetric Dirichlet priors Dirichlet$(\alpha, \alpha, \ldots, \alpha)$ over all simplices. While it simultaneously controls the parameters of the $\varphi$ and $\pi'$ matrix, a choice which is may not be often desired in practice, we hope that it provides meaningful information on the sensitivity of the inference to the choice of the prior. For $\alpha = 1$ this prior is uniform and has been studied in Sec. 4. Choosing $\alpha < 1$ encourages more sparsity: the prior on the $\pi'$ vector concentrates near to the boundary of the simplex (i.e., $(1, 0)$ and $(0, 1)$ vectors). Simultaneously, the model assumes that $\varphi$ matrix corresponds to sharper predictions. For example, this means that one of the entries $\{P(C = 1 \mid Y = 1), P(C = 2 \mid Y = 1)\}$ should be larger than the other, i.e., the true positive rate and false negative rate should be very different. For $\alpha > 1$ the prior distributes the mass around the center, preferring balanced prevalence vectors $\pi'$. Under this prior, the false negative rate and the true positive rate are similar.

We changed $\alpha \in \{0.1, 1, 10\}$ and performed Bayesian inference. Similarly as in Appendices E.1 and E.4 we ran four Markov chains with 500 warm-up steps and 1,000 samples each. To flag potential convergence issues, we calculated the potential scale reduction factors $\hat{R}$, which did not exceed 1.02. The minimal effective sample size was 362.

The posterior of different models is visualised in Fig. 12. We see that for a small sample size ($N = N' = 50$), the sparse prior ($\alpha = 0.1$) puts more mass at the boundary. At the same time, the predictions under $\alpha = 1$

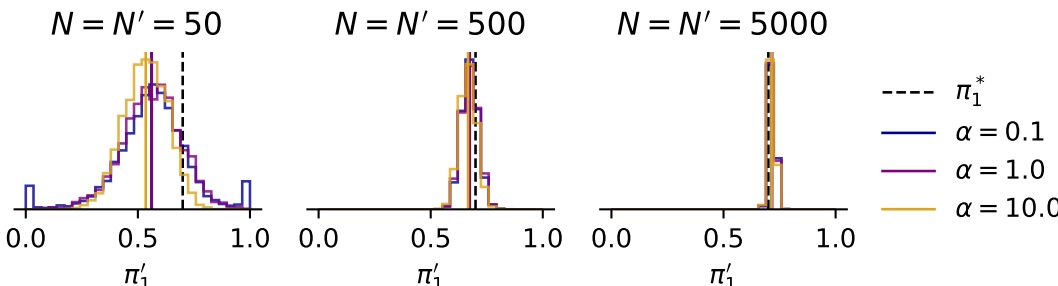

Figure 12: Sensitivity of the Bayesian inference with respect to the Dirichlet prior with concentration parameter $\alpha$ under different sample sizes $N = N'$. We plot the posterior in the form of a histogram, together with a vertical line representing posterior mean. The dashed line represents the ground-truth value.

and $\alpha = 10$ priors are nearly identical. For larger $N = N'$ the effect of the prior vanishes, with the posterior concentrating around the true value, which is in line with Theorem 3.1.

