# OpenReview forum: "Bayesian Quantification with Black-Box Estimators"
_TMLR — Accepted by TMLR_

### Review · Reviewer_StyW · 2024-04-12

**Summary Of Contributions:**

This paper studies the quantification problem and generalizes the point estimate methods like the black-box shift estimators to the Bayesian setting. The authors theoretically analyze the asymptotical consistency of the proposed maximum a posterior inference under weak assumptions. In particular, the proposed approach explicitly measures the uncertainty and does not suffer from the negative value problem in the low-data regime. Empirical evaluation on synthetic and real-world datasets shows that the proposal is competitive with existing non-Bayesian quantification methods.

**Audience:**

Yes

**Claims And Evidence:**

Yes

**Requested Changes:**

- On the presentation side, the authors might want to provide a more coherent description of the proposed method, highlighting the connections and the differences from existing method and reasons why it has the ability to resolve issues of existing methods. It also would be nice to have some comparisons between the "weak assumptions" used in Theorem 3.1 and the literature. Moreover, more detailed description of the full efficient Markov chain Monte Carlo sampling would be much appreciated to be included in Section 3.2.
- On the experimental side, the authors might want to 1) group the experiments according to the improvements of the proposed method and provide some discussion about the performance comparisons; 2) give clear definition of "weak uniform priors" used in the experiment and potentially some additional experiments to demonstrate the sensitivity to the different prior specifications; 3) include experiments used in the existing literature so that the audience could have a direct comparison; 4) present some summary quantitative metrics since the lines corresponds to different methods shown in Figure 3 and 5 are difficult to differentiate.
- In Section 5, the authors "stress the importance of a principal shift in perspective" about treating f as an auxiliary feature extraction method. But since this has been introduced in the literature especially the black-box shift estimator method, the authors might want to give a clearer argument regarding additional insight/perspective advocated in this work.
- Some notations: 1) "lab"/"unl" looks ambiguous to me, might consider "source"/"target" or just "train"/"test"; 2) it should be $C_{j}^{'}$ instead of $X_{j}^{'}$ in equation (3).

**Strengths And Weaknesses:**

*Strengths*:
- This paper addresses an interesting problem - the quantification problem which tries to estimate the class prevalence under the prior shift.
- This paper carefully analyzes the connections and pros and cons among different approaches to the quantification problem, and extends the black-box shift estimator to the Bayesian setting which mitigates several existing issues.
- The authors have demonstrated the claimed improvement including the robustness and superior performance in low-data scenarios through empirical evaluations.

*Weaknesses*:
- Overall, the paper is not easy to follow, and the presentation could be improved.
- The experiments could be more thorough and include datasets that are used in the literature. Please see Requested Changes for details.

---

> ### Author Response · Authors · 2024-04-20
> **Response**
>
> We would like to thank you for the very detailed review and precise suggestions on how to improve the clarity of the manuscript.
>
> We have introduced the following changes:
>
>   - We have fixed the typo in Eq. (3) and changed the notation from $P_\text{lab}$ and $P_\text{unl}$ to $P_\text{train}$ and $P_\text{test}$.
>   - Revised the description of the proposed method in Section 3.1 as suggested.
>   - We have adjusted Section 3.2, to clarify that we aim at removing the $N+N'$ factor from the likelihood evaluation, rather than proposing a new Hamiltonian MCMC sampling scheme.
>   - We do agree that the claim in Section 5 was not worded appropriately; we have rephrased it.
>   - We have provided more information on the priors used and added experiments investigating sensitivity to the prior choice (Appendix E.5).
>
> At the same time, we consider Fig. 3 and Fig. 5 only as simple summaries presenting the evidence that the proposed method is on par with the existing solutions. Although having seen a clear improvement in these settings would be an additional advantage of the proposed method, we interpret these results as confirming that existing point estimators are already of very good quality when problems are identifiable and data are sufficiently abundant.
> At the same time, there exists no standard set of experiments on which the quantification estimators are tested – for this reason we decided to conduct an extensive set of simulated experiments in Section 4.2.
>
> We would be gratful for any further comments regarding the introduced or further possible improvements.

---

### Review · Reviewer_qVtw · 2024-04-16

**Summary Of Contributions:**

This paper formulated the quantification problem as a Bayesian inference problem, and proposed a Bayesian approach that utilizes blackbox feature extractor and allows exact uncertainty quantification, which is particularly favorable in small sample scenario.

**Audience:**

Yes

**Broader Impact Concerns:**

The main contributions of the paper is on the methodology so to the best of my understanding, there is no major potential broader impact concerns.

**Claims And Evidence:**

Yes

**Requested Changes:**

Critical questions to clarify in the paper:

- How robust is the inference to the prior specification? When there is limited prior knowledge, what are some recommendations on the prior?
- How to choose the blackbox feature extractor f in practice?


Minor changes / questions:
- Clarify the notation P_lab and P_unl
- Theorem 3.1 proved some desirable properties of MAP. Then why is the posterior mean instead of MAP used as the point estimate in Section 4.2?

**Strengths And Weaknesses:**

Strength:

Clear problem formulation and motivation.

The Bayesian approach is natural and provides a principled way for uncertainty quantification of the estimated P_test(Y)

The proposed method is assessed and demonstrated from various aspects with both simulated data and scRNA data.

Weakness:
For the proposed method to be practically useful, there seems to be a lack of discussion on the choice of prior and feature extractor.

---

> ### Author Response · Authors · 2024-04-20
> **Response**
>
> Thank you very much for your review. We attach a revised draft of the paper, with annotated changes in the text. To summarize:
>
> - We changed the notation $P_\text{lab}$ and $P_\text{unl}$ to $P_\text{train}$ and $P_\text{test}$.
> - For the prior specification we now advise to the uniform prior (when no specific domain knowledge about the problem is available) and refer to a section in Gelman et al. (2013) discussing weakly informative priors.
> - We have added a small experiment (Appendix E.5) to check robustness to the prior choice.
> - We have rephrased the Discussion section to mention the problem of selecting $f$ in the cases when a generic one is not provided and one has enough data to train a new one.
>
> The question why to prefer posterior mean over the MAP estimate is an excellent one. The short answer would be the following:
>
>   - The whole posterior is preferable over any point estimate. We use point estimates only to compare with methods which do not offer the whole posterior. The mean estimate seems to be a good choice, compared to other methods.
>   - For non-identifiable problem settings (such as $K < L$) or small sample sizes, the mode of the distribution may not be unique, so that the MAP estimate is not well-defined.
>   - The mean can be calculated using the MCMC samples, while the MAP estimate requires optimization, rather than sampling.
>
> However, there is also a long answer, rooted in Bayesian decision theory (Bernardo and Smith, 1994). We decided to write a new section (Appendix C.1), explaining this argument.
>
> We would be very grateful for any further comments regarding the introduced changes.
>
> **References**
> - J.M. Bernardo and A.F.M. Smith (1994), *Bayesian Theory*, ISBN:9780471494645
> - A. Gelman et al. (2013), *Bayesian Data Analysis*, ISBN:9781439840955

---

### Review · Reviewer_6Rrn · 2024-04-17

**Summary Of Contributions:**

This is a paper well suited to TMLR since the focus is on "technical
correctness over subjective significance" [jmlr.org/tmlr]. A problem
we might care about ("quantification") is addressed competently, and
the method is described reasonably clearly. Related work and competing
methods are described properly. I am not sufficiently well versed in
this problem to be sure that the competing approaches are the SOTA, so
will have to trust the authors on this. The authors are careful with
their claims, including those concerning their proposed method. The
word "may" appears very frequently.

**Audience:**

Yes

**Claims And Evidence:**

Yes

**Requested Changes:**

1. Report on the convergence status of MCMC runs
2. Do at least one experiment on a sufficiently large instance to cause computational problems for the presented method (or prove that no such instance exists).

**Strengths And Weaknesses:**

As a result of taking a Bayesian approach to the problem at hand, this
paper has a pleasing simplicity. The assumed model is conveniently
represented as a DAG, and "learning" reduces to finding (or
approximating) a posterior distribution over the unknown(s) of
interest. The advance over the Bayesian approach (to the same problem)
of Storkey is that it does not rely on a generative model
P(X|Y). Whether a generative mechanism for (PX|Y) might be
misspecified is highly contingent: presumably one could have real
situations where it is possible to specify it reasonably correctly. As
the authors correctly state: "M_{approx} loses information from the
problem, so we do not expect it to be as appropriate as a properly
specified generative model for P(X|Y)."

The authors "work with a given dimension reduction mapping", and, as
they explain, much depends on the appropriateness of the choice of
mapping and whether it leads to a sufficient statistic (as the authors
do explain).

A motivation for dimensionality reduction (and thus losing
information) is that: "However, for high-dimensional $\theta$, or when
N and N' are large, MCMC will generally not be tractable". This is a
vague statement about using MCMC for this problem; it would be good to
see evidence of problems with MCMC (by eg citing some paper showing
it, or doing the experiments yourselves). The statement: "We did not
compare obtained posteriors in high-dimensional problems due to high
computational costs associated with running MCMC on high-dimensional
generative models." is not evidence.

It is true that $\tilde{K}_{y}(.;\phi)$ (and thus M_approx) is
particularly simple which is indeed an advantage of the presented
method. In Section 3.2 the categorical distributions are effectively
replaced with multinomial ones, which makes sense.

The experiments are fairly small. It would be good to see what happens
with large problems - and to see where, if anywhere, we hit the limits
of scalability for the presented method. Also, there is no discussion
of convergence (or lack thereof) of the MCMC samplers. I see that the
authors use numpyro to do the MCMC; does this package not report, eg
rhat values?

MINOR POINTS

The "panels" in Fig 3 are individual sub-figures. I recommend just
labelling them as 3(a) - 3(f) and refer to them thus (rather than "the
first panel of the second row", etc). The sixth "panel" (concerning
model misspecification) is never explicitly referenced in the text.

"To formalize the problem, consider a probabilistic graphical model in
Fig. 1" State that the model referred to is M_true, rather than
requiring the reader to deduce this from the text that follows this
statement.

---

> ### Author Response · Authors · 2024-04-20
> **Response**
>
> We would like to thank you for a detailed review of our manuscript. We especially appreciate the comment that "As a result of taking a Bayesian approach to the problem at hand, this paper has a pleasing simplicity."
>
> **Minor changes**
> We have adjusted Fig. 3, so that now we refer to individual subfigures, and included an in-text reference to $\mathcal{M}_\text{true}$.
>
> **MCMC convergence**
> Regarding the MCMC runs, we have:
>   - Added to the Appendix the information about $\hat R$ and effective sample size for the experiments described in Sections 4.1 and 4.4.
>   - Added a clarification that we did not check $\hat R$ for the experiments described in Sections 4.2 and 4.3. As these experiments describe the frequency properties of the proposed Bayesian estimator, these experiments required considerably more computational resources than the other experiments and we decided to run only one chain (rather than four) for these experiments. As we used the settings from Sections 4.1 and 4.4 and the performance of the method seemed to be good, we did hope that the chains were reasonably close to the stationary distribution.
>
> We now clarify these experimental details in the appendix.
>
> **Computational issues and tractability of MCMC**
> We do agree that the statement on MCMC tractability in the text was not precise enough; we have changed it and added the references discussing possible computational issues with running MCMC on large data sets and high-dimensional problems.
>
> We consider this a limitation also for our method: so the question whether there exist situations where our method will suffer from computational issues has a confirmatory answer. Namely, to sample $\varphi$ and $\pi'$ one has to explore a space with $O(KL)$ dimensions. For $K\ge L$ and large data sets (such as ImageNet with $L\ge 5\cdot 10^3$), this space has millions of dimensions and running MCMC on such high-dimensional problems is known to be computationally demanding.
> We have added this limitation to the Discussion, together with appropriate references.
>
> We would be very grateful for your opinion on these changes.

---

> > ### Comment · Reviewer_6Rrn · 2024-04-27
> >
> > I'm not happy about doing MCMC with only a single chain (Appendix E.2). This means we don't have evidence about convergence and so no evidence that the MCMC sample provides a reasonable approximation to the desired posterior. I understand that these are computationally intensive runs, but it's the machines that do the work (while you sleep perhaps), so one just needs patience.
> >
> > More generally, I now notice that I can't find details on how long (in seconds) the various MCMC runs took. Of course, this depends on machine used, implementation etc but it is useful for a reader to get some idea of how much compute resource they would need to tackle the problems addressed in this paper. Typically ML papers briefly provide info on machine used (including CPU MHz, RAM etc) and how algorithms were implemented but this is not give here, which it should be.
> >
> > The request in my review was to see **where** not **whether** serious scalability problems occur but at least we have a prediction in the Discussion as to when one might hit computational problems.

---

### Author Response · Authors · 2024-04-20
**General response**

We would like to thank all Reviewers for their thorough reviews and many interesting suggestions.

We have uploaded a revised version of the manuscript addressing the following points:

- We adjusted the notation and revised the description of the proposed method. We discuss now the prior choice as well as the assumptions of Theorem 3.1.
- We improved on the descriptions of the experiments. Additional information on MCMC runs is now provided in the appendix.
- We discuss the prior specification in more detail. Additionally, we added Appendix E.5, investigating the sensitivity to a prior choice in a simulated data set.
- We revised the Discussion as suggested.

Please, note that this version is not final and was prepared to show the revisions made.

---

### Comment · Action_Editor_EsUx · 2024-04-24
**Discussion period**

Dear reviewers,

Thank you so much for submitting detailed and thoughtful reviews! Please take a moment to read the other reviews as well as the responses from the authors, and request any additional information you may need for making a decision recommendation.

---

### Decision · Action_Editor_EsUx · 2024-05-22

**Recommendation:** Accept with minor revision

**Comment:**

The paper presents a sound and interesting approach to the quantification problem. The methodology is reasonable, and the authors provide sufficient support for the claims made in the paper. The topic is relevant to the field of distribution shifts, as well as statistics. Thus, the paper meets both the evidence and the audience criteria of TMLR.

All reviewers are unanimously leaning towards acceptance. The consensus is that the paper is sound and provides a meaningful contribution to TMLR. During the discussion period, the authors addressed almost all of the concerns raised by the reviewers.

**Requested revision**: Please address the remaining questions of reviewer 6Rrn ([link](https://openreview.net/forum?id=Ft4kHrOawZ&noteId=ZNteGLwnus)). In particular, please describe the compute necessary for running the methods. Ideally, please run the requested missing experiments with multiple MCMC chains and $\hat R$ estimates, or provide a discussion of why it is impractical to run these experiments.

**Audience:**

The paper provides a Bayesian approach to tackling the problem of estimating the class distribution of unlabeled examples. This is a meaningful problem, and the paper could be interesting to the TMLR audience, in particular to those interested in statistics or robustness to distribution shifts.

**Claims And Evidence:**

The paper studies the problem of _quantification_: estimating the distribution of classes in a test set, given the training data $(X_i, Y_i)$ and unlabeled test data $X_j'$. The authors approach this problem via Bayesian inference, and also introduce a dimensionality reduction for more efficient inference. The authors show competitive results with established methods in a variety of experiments.

All of the reviewers agree that the claims in the paper are generally sound and well-supported by evidence.
The only missing evidence is support for convergence of the MCMC chains in the experiments, requested by reviewer 6Rrn ([link](https://openreview.net/forum?id=Ft4kHrOawZ&noteId=pKXcT7hOd8)).

---

> ### Author Response · Authors · 2024-06-04
> **Camera-ready version**
>
> We would like to once again thank the Reviewers and the Action Editor for many valuable suggestions. Basing on your feedback, we have made the following changes:
>
> - All experiments now report $\hat R$.
> - Computing requirements are described in Appendix E.
> - Additional proofreading has been completed to improve clarity and readability.
>
> The camera-ready version has been uploaded to OpenReview.
> Thank you for your time and consideration.